# Analysis of Environmental Factors Associated with Cyanobacterial Dominance after River Weir Installation

**Sungjin Kim [1], Sewoong Chung [1,*] , Hyungseok Park [1], Youngcheol Cho [1] and Heesuk Lee [2]**

[1]  Department of Environmental Engineering, Chungbuk National University, Cheongju 28644, Korea; znssungjin2@naver.com (S.K.); qwrs07@chungbuk.ac.kr (H.P.); choy@chungbuk.ac.kr (Y.C.)

[2]  K-water; Daejeon 34350, Korea; heesuklee@gmail.com

*  Correspondence: schung@chungbuk.ac.kr; Tel.: +82-43-261-3370

**Abstract:** Following the installation of 16 weirs in South Korea's major rivers through the Four Rivers Project (2010–2012), the water residence time increased significantly. Accordingly, cyanobacterial blooms have occurred frequently, raising concerns regarding water use and the aquatic ecosystem health. This study analyzed the environmental factors associated with cyanobacterial dominance at four weirs on the Nakdong River through field measurements, and parametric and non-parametric data mining methods. The environmental factors related to cyanobacterial dominance were the seven-day cumulative rainfall (APRCP7), seven-day averaged flow (Q7day), water temperature (Temp), stratification strength ($\Delta T$), electrical conductivity (EC), dissolved oxygen (DO), pH, and $NO_3-N$, $NH_3-N$, total nitrogen (TN), total phosphorous (TP), $PO_4-P$, chlorophyll–a, Fe, total organic carbon (TOC), and $SiO_2$ content, along with biological and chemical oxygen demands. The results indicate that site-specific environmental factors contributed to the cyanobacterial dominance for each weir. In general, the physical characteristics of EC, APRCP7, Q7day, Temp, and $\Delta T$ were the most important factors influencing cyanobacterial dominance. The EC was strongly associated with cyanobacterial dominance at the weirs because high EC indicated persistent low flow conditions. A minor correlation was obtained between nutrients and cyanobacterial dominance in all but one of the weirs. The results provide valuable information regarding the effective countermeasures against cyanobacterial overgrowth in rivers.

**Keywords:** cyanobacterial dominance; data mining; environmental factors; Four Rivers Project; weirs

## 1. Introduction

Abnormal overgrowth of cyanobacteria (hereinafter referred to as "algal blooms") in rivers and lakes adversely affects the biodiversity in freshwater ecosystems [1] and affects water usage by generating taste and odor compounds in drinking water [2,3]. Furthermore, the mass proliferation of harmful cyanobacteria, which generates toxins, can prove fatal to both livestock and humans [4,5]. In the past, algal blooms in South Korea have caused sporadic problems in stagnant waters such as those of dam reservoirs and estuary lakes. However, since 16 weirs were installed on major rivers as part of the Four Rivers Project (2010–2012), the algal blooms have been occurring widely in large rivers such as the Nakdong, Geum, and Yeongsan Rivers, causing a major socio-environmental issue [6].

There are various opinions on the cause of the algal blooms that frequently occur at weirs installed on major rivers [7,8]. Intensive studies have been performed in the lowland rivers of south-eastern Australia such as the Murrumbidgee [9,10], the Murray [11], and the Barwon-Darling [12,13] to investigate the effect of weirs on cyanobacterial blooms. The results suggest that the overgrowth of

cyanobacteria is closely linked to low river discharge. Progressions from diatoms to cyanobacteria as dominant phytoplankton species were found to be dependent on the formation of persistent thermal stratification [10].

However, universal interpretation is difficult because the causes of algal blooms vary with the characteristics of a given watershed and water body, and because of the complex interactions between various factors. There are many different causes of algal blooms following excessive proliferation of cyanobacteria, which are largely divided into physical, biogeochemical, and physiological factors [4]. In terms of physical factors, cyanobacteria exhibit optimal growth at high temperatures (i.e., above 20 °C) and occur in large quantities in water bodies with low flow velocity and long residence time [14,15]. Cyanobacterial outbreaks coincide with (or are strongly associated with) the occurrence of thermal stratification [5,10,11,13,16,17], where the upper and lower layers of water bodies are not mixed and the water temperature in the upper layer remains high, which is appropriate for cyanobacterial growth. Furthermore, as the vertical turbulent mixing weakens, diatoms and green algae (with no buoyancy control function) sink below the photic zone, whereas cyanobacteria tend to be concentrated in the surface layer [18,19]. Chung et al. [20] demonstrated that the overgrowth and spatiotemporal distribution characteristics of *Microcystis* in a reservoir with strong thermal stratification are determined by the interaction between the physical mixing characteristics in the reservoir and the buoyancy control function of cyanobacteria. Furthermore, Moss and Balls [21] and Wu et al. [22] analyzed phytoplankton production using hydrological factors such as precipitation, flow velocity, and channel width. Their results showed that the phytoplankton community structure in the studied rivers correlated significantly with physical factors such as channel width and flow velocity.

In terms of biogeochemical factors, cyanobacterial growth is determined by the concentrations of essential nutrients such as phosphorous (P) and nitrogen (N) in conditions where physical factors (e.g., water temperature, quantity of light, and residence time) are satisfied. In most freshwater ecosystems, P acts as a growth-limiting nutrient for algae [23–25]. Previously, Ahn et al. [25] argued that with sufficient solar radiation, cyanobacterial growth is determined by the N/P concentration ratio and cyanobacteria proliferation can be controlled by inducing the conversion of toxic cyanobacteria to other non-toxic algae species by changing the N/P ratio in the water. The P supply in a water body is dependent upon external sources, i.e., point and non-point sources in the watershed, and the internal source of the water-body sedimentary layer. The release of P from the sedimentary layer occurs when the oxygen (O) supply to the lower layer is limited and the sediment oxygen demand is high in stratified water layers causing the water to become anaerobic; these conditions can also promote cyanobacteria proliferation [26,27]. Furthermore, micronutrients such as iron (Fe) and molybdenum (Mo) also have considerable effects on cyanobacterial growth [28]; Mo affects N absorption and carbon (C) fixation, whereas Fe affects photosynthesis and N fixation. Furthermore, the pH and alkalinity influence changes in the inorganic C sources ($CO_2$, $CO_3^{2-}$, and $HCO_3^-$) of cyanobacteria and affect their growth [5,29,30]. The pH has a dominant influence on the various forms of dissolved inorganic carbons ($CO_3^{2-}$, $HCO_3^-$, and $H_2CO_3$) employed in photosynthesis, and also has a high correlation with the C sources of phytoplankton [31–33].

Regarding physiological factors, cyanobacteria move vertically in a water body through buoyancy control using intracellular gas vesicles [34,35]. The buoyancy mechanisms of cyanobacterial species such as *Microcystis* and *Oscillatoria* are important factors influencing cyanobacterial outbreaks [36–38]. Furthermore, cyanobacteria form spherical colonies, filaments, and trichomes surrounded by mucus and have less zooplankton predation than other algae [39,40]. Some cyanobacteria such as *Anabaena* are not affected by N limitations because they can fix N in the air [41]. In addition, they communicate with each other using specific chemicals and adjust various collective metabolic activities for survival in a specific environment, depending on whether the cyanobacteria are experiencing a period of increase or decrease [42–45].

Thus, there are many different causes of algal blooms and the causal relationships have too much uncertainty to be analyzed through a mechanical modeling method. This uncertainty arises

from the existence of highly complex factors such as the watershed pollutant load, seasonal changes in weather and hydrological conditions, thermal stratification and hydraulic mixing characteristics, spatiotemporal nutrient distribution, and physiological characteristics of emerging species. Therefore, various statistical methods and data modeling techniques are applied to evaluate the importance of the various causes of algal blooms [46–49]. Cho et al. [31] analyzed the causes of algal blooms by conducting a cluster analysis of the physicochemical factors using a self-organizing map (SOM) for 12 lakes in the southwestern part of South Korea. Their results indicate that physical factors such as water temperature have greater effects on phytoplankton cluster changes than chemical factors such as nutrients for growth [31]. According to Horne and Goldman [50] and Sze [51], phytoplankton clusters exhibit low correlations with physicochemical factors. In the Asian monsoon climate region, increases in phytoplankton populations in lakes are more strongly affected by meteorological and hydrological factors such as rainfall and flow rate than by physicochemical factors [52].

For the mid-stream of the Nakdong River, which is the subject river of this study, Jung and Kim [53] analyzed the relationship between chlorophyll a (Chl–a) and nutrients through the application of multivariate statistical methods (such as principal component and factor analysis), and reported that, in summer, algae are more affected by P- than N-based factors. However, the correlations between the Chl–a concentration, N, P, and other factors at each point along the river were different, likely due to the fact that the N, P, and Chl–a concentrations are affected by the pollution load, turbidity increase, and residence time variation due to rainfall runoff in rivers [53]. Therefore, for accurate analysis of the factors associated with algal overgrowth in rivers, additional data analysis of the meteorological and hydrological factors (such as rainfall and flow rates) are necessary, rather than the simple use of water quality as an explanatory variable [52]. Although many studies on the causes of algal blooms have been conducted, there is insufficient research on the causes of such blooms in large rivers, in which the physicochemical environment changes rapidly according to variations in precipitation and flow rate. Consequently, analysis of the correlations between the physicochemical factors influencing cyanobacterial dominance (which is the cause of algal blooms) through data mining is critical. In addition, comprehensive analyses of the important variables are essential for algal bloom control.

In this study, an intensive field survey and laboratory measurements were conducted on four weirs of the Nakdong River, where algal blooms due to cyanobacteria have occurred frequently since the installation of these weirs in 2012. The environmental conditions of each weir were also investigated to determine their influence on the cyanobacterial dominance and to assess the major variables for controlling cyanobacteria. This was achieved by applying statistical analysis and data modeling techniques for collected weather, hydrological, water quality, and algal data. The qualitative and quantitative measurements of the water quality and phytoplankton of each target weir were performed from May 2017 to November 2018. In addition, a correlation analysis between the cyanobacteria cell density and environmental factors was conducted, along with a variable importance evaluation using step-wise multiple linear regression (SMLR) and random forest (RF) models. Furthermore, the environmental factors of cyanobacterial dominance for each weir were analyzed based on the results of various parametric and non-parametric data mining methods, such as recursive feature elimination using RF (RFE-RF), decision tree (DTs), and principal component analysis (PCA).

## 2. Materials and Methods

### 2.1. Site Description

The target sites of this study were the Gangjeong Goryeong (GGW), Dalseong (DSW), Hapcheon Changnyeong (HCW), and Changnyeong Haman (CHW) weirs on the Nakdong River, South Korea. These weirs are among the eight weirs installed on the Nakdong River as part of the Four Rivers Project (Figure 1) and are located in the midstream and downstream regions. The GGW, DSW, HCW, and CHW watershed areas (i.e., total water storage capacities) are 11,667 (92 million $m^3$), 14,248 (59 million $m^3$), 15,074 (70 million $m^3$), and 20,697 $km^2$ (101 million $m^3$), respectively [54]. The CHW

is the most downstream weir of the Nakdong River. Since the weirs were completed in 2012, algal blooms have occurred frequently due to cyanobacteria overgrowth in summer. The highest level of the harmful cyanobacteria cell density guideline of the World Health Organization (WHO), i.e., 20,000 cells mL$^{-1}$, has been exceeded 36 (peak: 261,219 cells mL$^{-1}$), 69 (peak: 495,360 cells mL$^{-1}$), 95 (peak: 1,264,052 cells mL$^{-1}$), and 80 times (peak: 715,993 cells mL$^{-1}$) at GGW, DSW, HCW, and CHW respectively [55]. The overgrowth of harmful cyanobacteria in the Nakdong River has emerged as a social issue because this river is an important water source for 13 million citizens living in the Daegu, Busan, and Gyeongnam regions.

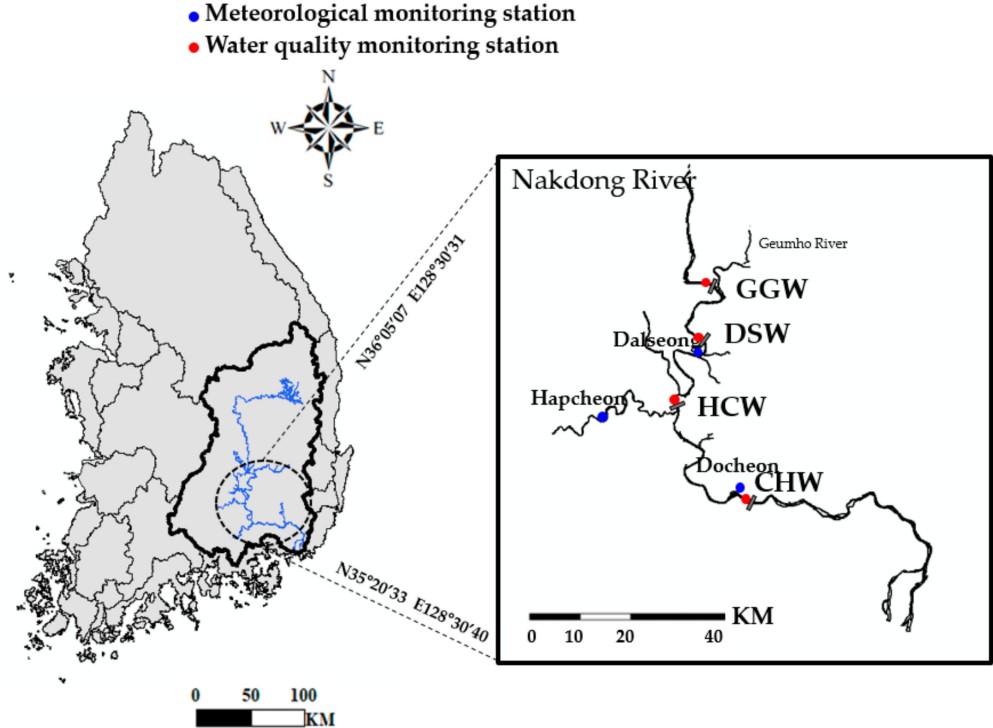

**Figure 1.** Locations of study weirs and sampling stations; Gangjeong Goryeong weir (GGW), Dalseong weir (DSW), Hapcheon Changnyeong weir (HCW), and Changnyeong Haman weir (CHW).

*2.2. Sampling and Analysis*

The samples were collected at the locations of regular water quality monitoring stations operated by the Ministry of Environment of Korea, 500 m upstream of the four study weirs, between May 16, 2017, and November 23, 2018. The samples were collected 36 times at GSW and DSW, respectively (108 samples); 32 times at HCW (96 samples); and 31 times at CHW (93 samples). The samples were collected from the top, middle, and bottom layers in the middle of the river using a Van Dorn water sampler. The water temperature (Temp, °C), pH, dissolved oxygen (DO, mg L$^{-1}$), and electrical conductivity (EC, μS cm$^{-1}$) were measured at different depths at each site using a multi-item water quality meter (YSI-EXO, YSI-6600, YSI Pro plus). The pH, DO, and EC sensors were calibrated every week. The collected samples were stored at ≤ 4 °C and transported to the laboratory, where the properties (excluding water temperature, DO, EC, and pH) were assessed in accordance with the Standard Methods for the Examination of Water Pollution [56].

For analysis of the number of algal species, the algae in the samples were fixed with Lugol solution on site, before being transported to the laboratory. They were analyzed in accordance with the "Phytoplankton-Microscope Counting Method (ES 04705.1b)" of the Standard Methods for the Examination of Water Pollution [56]. The *Microcystis*, however, exhibited uneven distributions that generated counting errors; thus, *Microcystis* colonies were separated from a certain amount of solution and counted separately.

The meteorological data were collected from the general and disaster prevention meteorological monitoring stations located in the project area; these data are provided on the meteorological data open portal of the Meteorological Administration [57]. The precipitation data (mm) were collected from the four study sites of the Nakdong River, and the flow rate data were collected from the hydrological data of each weir on the K-water information portal [58].

## 2.3. Statistical Analyses

The research procedure implemented in this study is illustrated in Figure 2. In the final step, the cyanobacterial dominance environment of each weir was comprehensively evaluated. Furthermore, the correlations of the environmental factors with the cyanobacteria (Cyano), green algae (Green), and diatom (Diatom) cell densities, and Chl–a concentration were analyzed following the classification of the collected data into field measurements, physical factors, nutrients, and organic matter. The field measurements included Temp, DO, and EC. The physical factors were the seven-day cumulative rainfall (APRCP7), seven-day averaged flow (Q7day), and the water temperature difference between the top and bottom layers. The nutrients were the total P (TP), total N (TN), $NH_3$–N, and $NO_3$–N. The organic matter and trace materials included the biological and chemical oxygen demands (BOD and COD, respectively), total organic C (TOC), Fe, and $SiO_2$.

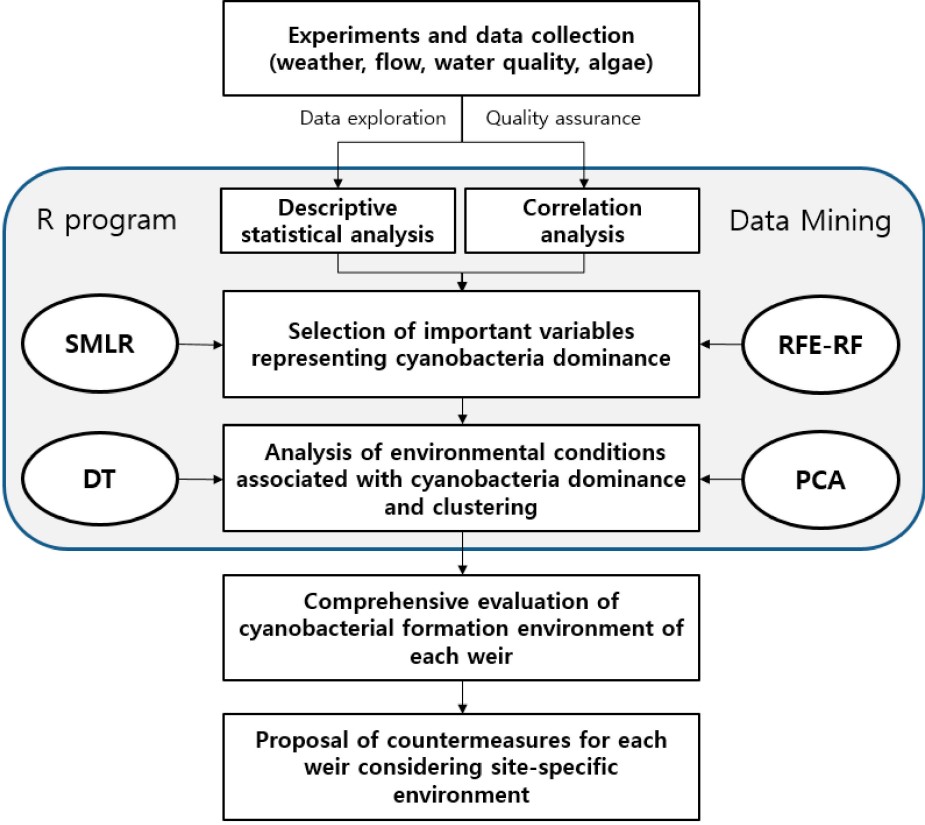

**Figure 2.** Flowchart of statistical analyses and data mining processes employed in this study. SMLR - step-wise multiple linear regression, RFE-RF - recursive feature elimination using random forest models, DT- decision tree, PCA – principle component analysis.

The SMLR model was used as the parametric method for evaluating the important variables of the cyanobacteria-dominated environment. The dependent variable used in this analysis was the cyanobacteria dominance (C.dominance) and the independent variables were Temp, DO, EC, Q7day, APRCP7, ΔT, pH, $NO_3$–N, $NH_3$–N, TN, $PO_4$–P, and Fe. The C.dominance represents the ratio of the cyanobacteria cell density to the total algae, indirectly representing the risk of algal blooms due to

cyanobacterial dominance. The SMLR model excludes independent variables with low statistical significance in a step-by-step manner to create the multiple linear regression model with the best prediction performance. In this study, the stepwise regression method with forward selection was applied to the SMLR. For the first variable used in the model, the independent variable with the largest positive or negative correlation with the dependent variable was selected, and the independent variables with strong correlations were applied sequentially. The procedure was terminated if no variable satisfied the entry criteria [20]. For the model evaluation, the adjusted coefficient of determination (Adj. $R^2$), root mean square error (RMSE), Mallows' $C_P$ statistic, and Akaike information criterion (AIC) were used. The SMLR results were used to determine the important variables to be applied to the selection of a parsimonious multiple regression model.

The RFE-RF model was used in the non-parametric method for evaluating the importance of the environmental variables with regard to cyanobacterial dominance; this is an ensemble learning method. Recursive feature elimination is a backward selection method for identifying a parsimonious model that achieves the maximum prediction performance using the minimum variables, and in which variables with low importance are removed one by one. The environmental conditions corresponding to high C.dominance were found using the important variables selected as DT input variables. The dependent variable of the analysis was C.dominance and the explanatory variables were Temp, DO, EC, Q7day, APRCP7, $\triangle$T, pH, $NO_3$–N, $NH_3$–N, TN, $PO_4$–P, and Fe.

For the DT model analysis, the rpart package [59] of the R program was used. For the analysis of the RFE-RF model, the caret [60] and randomForest [61] R packages were used. The DT model forms a tree model by repeatedly differentiating between each variable. The classification is performed if the dependent variable is categorical, and the regression analysis is performed if the dependent variable is continuous. The independent variables can be applied to both the categorical and continuous dependent variables. Following the repetition of this process, the final prediction model is selected through pruning to find an appropriate tree model.

The RF model is an ensemble learning method that yields improved accuracy by combining the predictions of several models for the given data after learning multiple models from the data. The RF prediction results are determined through the voting and averaging for classification and regression, respectively [59]. The RF model has the advantages of very high predictive power, model stability, and a lack of tuning parameters related to pruning and growth when there are multiple explanatory variables. In contrast, the researcher must select the number of trees (*ntree*) to be used to predict the target variable value and the number of explanatory variables (*mtry*) in each node of the trees. In this study, a default value of 500 was used for *ntree*, the main variable of the RF model. For *mtry*, the optimal value that provided the best prediction performance and minimized the out-of-bag (OOB) error was determined. For the RF model application process, the result was estimated after the exploratory data analysis (EDA) such as data collection, sorting, and missing-value processing. Furthermore, the *k*-fold cross-validation with *k* = 10 was performed three times to prevent over-fitting of the RF model and to evaluate the prediction performance during the development of the RFE-RF model.

An unsupervised learning method, PCA, was used for the cluster analysis of the environmental factors affecting algal bloom. The data used in this analysis were the C.dominance, Temp, DO, EC, Q7day, APRCP7, $\triangle$T, pH, $NO_3$–N, $NH_3$–N, TN, TP, $PO_4$–P, Chl–a, Fe, BOD, COD, TOC, and $SiO_2$. For determination of the number of principal components, only the principal component axes having eigenvalues of 1.0 or higher—which represent the variation of the data normalized to the principal component axis—were considered [62–64]. In PCA, the factor axis is rotated by simplifying the factor pattern structure to facilitate the factor analysis. The factor axis rotation methods include oblique and orthogonal rotation. The orthogonal rotation methods include Equimax, Varimax, and Quartimax [65]. The Varimax rotation method was applied in this study. Furthermore, to determine whether the raw data were appropriate for PCA, Bartlett's test of sphericity and the Kaiser-Meyer-Olkin (KMO) test were performed. Bartlett's test of sphericity is performed under the hypothesis that the correlation matrix of the data used for analysis is an identity matrix. When this hypothesis is validated ($p > 0.05$),

the data are inadequate for use in the PCA analysis. The KMO test result is a measure of the covariance among the factors inherent in the variables and data used for the analysis. The closer the result is to 1, the higher the analysis validity. The analysis can be performed only when this value is 0.5 or higher [63]. The analysis was performed after removing the variables with KMO values lower than 0.5 for each weir. The KMO test results showed that all the variables for HCW satisfied the KMO reference value (0.5 or higher). However, the KMO reference value was not satisfied by the Chl–a, BOD, $NH_3$–N, and DO variables (four variables) for GGW; by DO, Temp, $\triangle$T, $PO_4$–P, $NH_3$–N, Fe, $SiO_2$, $NO_3$–N, APRCP7, Q7day, and EC (11 variables) for DSW; and by Temp and TOC (two variables) for CHW. Thus, these variables were excluded from the analysis. From the analyses performed with the selected variables, all variables for all weirs satisfied the reference value (GGW: KMO value = 0.73, $p < 0.05$; DSW: KMO value = 0.57, $p < 0.05$; HCW: KMO value = 0.64, $p < 0.05$; and CHW: KMO value = 0.64, $p < 0.05$).

## 3. Results and Discussion

### 3.1. Descriptive Statistics of Collected Data

The descriptive statistics of the water-quality and algal data measured at the four study weirs and the flow data are outlined in Table 1. The mean pH of all four weirs were similar (8.1-8.2). The DO was 8.63–9.20 mg $L^{-1}$, the EC was 237.5–318 $\mu$S $cm^{-1}$, the SS was 8.66–10.1 mg $L^{-1}$, and the TN was 2.18–2.83 mg $L^{-1}$. The strongest thermal stratification during the survey period was recorded at GGW. The DSW exhibited the highest pollution with a BOD of 2.23 (±2.85) mg $L^{-1}$ indicating a high organic matter loading, and a TP of 0.087 (±0.1) mg $L^{-1}$ indicating high nutrient concentrations, due to the effect of the inflow from a polluted tributary, the Geumho River. The highest Chl–a concentration was measured at DSW, at 36.5 (±162.7) mg $m^{-3}$. The highest cyanobacterial cell densities were found at HCW and DSW at 10,761 (maximum: 453,283) cells $mL^{-1}$ and 9,236 (maximum: 694,667) cells $mL^{-1}$, respectively.

**Table 1.** Mean values (± standard deviation) of water quality data from the Gangjeong Goryeong weir (GGW), Dalseong weir (DSW), Hapcheon Changnyeong weir (HCW), and Changnyeong Haman weir (CHW). Note: see the main text for definitions of variables.

| Variable | Unit | Weir | | | |
|---|---|---|---|---|---|
| | | **GGW** | **DSW** | **HCW** | **CHW** |
| Sample size | *n* | 108 | 108 | 96 | 93 |
| Temperature | °C | 23.7 (±4.8) | 23.9 (±4.8) | 23.9 (±5.2) | 25.2 (±4.5) |
| pH | - | 8.2 (±0.6) | 8.2 (±0.8) | 8.1 (±0.8) | 8.2 (±0.8) |
| DO | mg $L^{-1}$ | 9.01 (±1.79) | 9.20 (±1.90) | 9.07 (±2.15) | 8.63 (±1.74) |
| EC | $\mu$S $cm^{-1}$ | 237.5 (±59.0) | 318.4 (±89.6) | 312.7 (±107.1) | 259.3 (±64.6) |
| SS | mg $L^{-1}$ | 8.66 (±7.86) | 10.1 (±13.7) | 8.79 (±6.44) | 9.80 (±5.55) |
| BOD | mg $L^{-1}$ | 1.91 (±0.88) | 2.23 (±2.85) | 1.98 (±1.22) | 2.08 (±1.02) |
| COD | mg $L^{-1}$ | 4.85 (±0.89) | 6.16 (±5.34) | 5.59 (±0.90) | 5.04 (±0.81) |
| TOC | mg $L^{-1}$ | 4.30 (±1.14) | 4.45 (±1.16) | 4.36 (±0.96) | 4.01 (±0.78) |
| TN | mg $L^{-1}$ | 2.29 (±0.48) | 2.83 (±0.85) | 2.61 (±0.50) | 2.18 (±0.41) |
| $NH_3$–N | mg $L^{-1}$ | 0.121 (±0.094) | 0.144 (±0.102) | 0.102 (±0.076) | 0.107 (±0.085) |
| $NO_3$–N | mg $L^{-1}$ | 1.65 (±0.43) | 2.03 (±0.50) | 1.88 (±0.50) | 1.48 (±0.46) |
| TP | mg $L^{-1}$ | 0.068 (±0.036) | 0.087 (±0.100) | 0.072 (±0.039) | 0.078 (±0.063) |
| $PO_4$–P | mg $L^{-1}$ | 0.037 (±0.069) | 0.035 (±0.033) | 0.027 (±0.021) | 0.028 (±0.022) |
| Chl–a | mg $m^{-3}$ | 14.7 (±11.3) | 36.5 (±162.7) | 24.0 (±21.0) | 26.2 (±15.6) |
| Cyano | cells $mL^{-1}$ | 3398 (±9,663) | 9236 (±67,068) | 10,761 (±48,555) | 7,323 (±25,767) |
| Green | cells $mL^{-1}$ | 3618 (±7,654) | 3239 (±3,037) | 3150 (±4035) | 3,250 (±3472) |
| Diatom | cells $mL^{-1}$ | 2582 (±2,395) | 3491 (±3,513) | 4139 (±3971) | 5,312 (±3847) |
| Outflow7 | $m^3$ $s^{-1}$ | 173.91 (±171.43) | 172.83 (±144.54) | 180.19 (±143.73) | 287.08 (±191.96) |
| APRCP7 | mm | 29.8 (±41.7) | 17.6 (±23.4) | 19.5 (±20.7) | 22.7 (±22.4) |
| $\triangle$T | °C | 2.2 (±1.7) | 1.5 (±1.1) | 1.0 (±1.2) | 0.6 (±0.8) |
| Fe | mg $L^{-1}$ | 0.10 (±0.06) | 0.01 (±0.01) | 0.08 (±0.05) | 0.07 (±0.04) |
| $SiO_2$ | mg $L^{-1}$ | 4.93 (±3.43) | 4.17 (±3.29) | 3.59 (±2.71) | 2.81 (±2.80) |

The examination of the cell density changes of each algal class in 2017 (Figure 3a) and 2018 (Figure 3b) revealed that the cyanobacterial dominance occurred at different times. In 2017, during the research period, cyanobacteria dominance was observed in all weirs in June, July, and early October; the dominant genus was *Aphanizomenon*. In 2018, cyanobacteria dominated from late July to August, and the dominant genus was *Microcystis*. During the research period, cyanobacterial dominance occurred mostly during periods of rainless days. This suggests that the transition characteristics of the dominant algae class in the four study weirs at the time of the study were closely related to the rainfall-runoff pattern. In 2017, rainless days persisted and the precipitation was very low in the period from April to June, whereas the precipitation was high in July and August. As a result, in 2017 the cyanobacteria began to dominate when the water temperature began to rise. The cyanobacteria disappeared in July and August, when there was frequent rainfall, and dominated in late September and October when persistent rainless days returned. However, in 2018, there were many rainy days in May and June and the precipitation was high. The rainy season began in late June and ended soon after in early July. As a result, the precipitation was low and heatwaves continued from mid-July to late August. Because of this phenomenon, the period of cyanobacterial dominance in 2018 was delayed compared to 2017, and the algal bloom began to appear from mid-July and disappeared when heavy rainfalls occurred in late August.

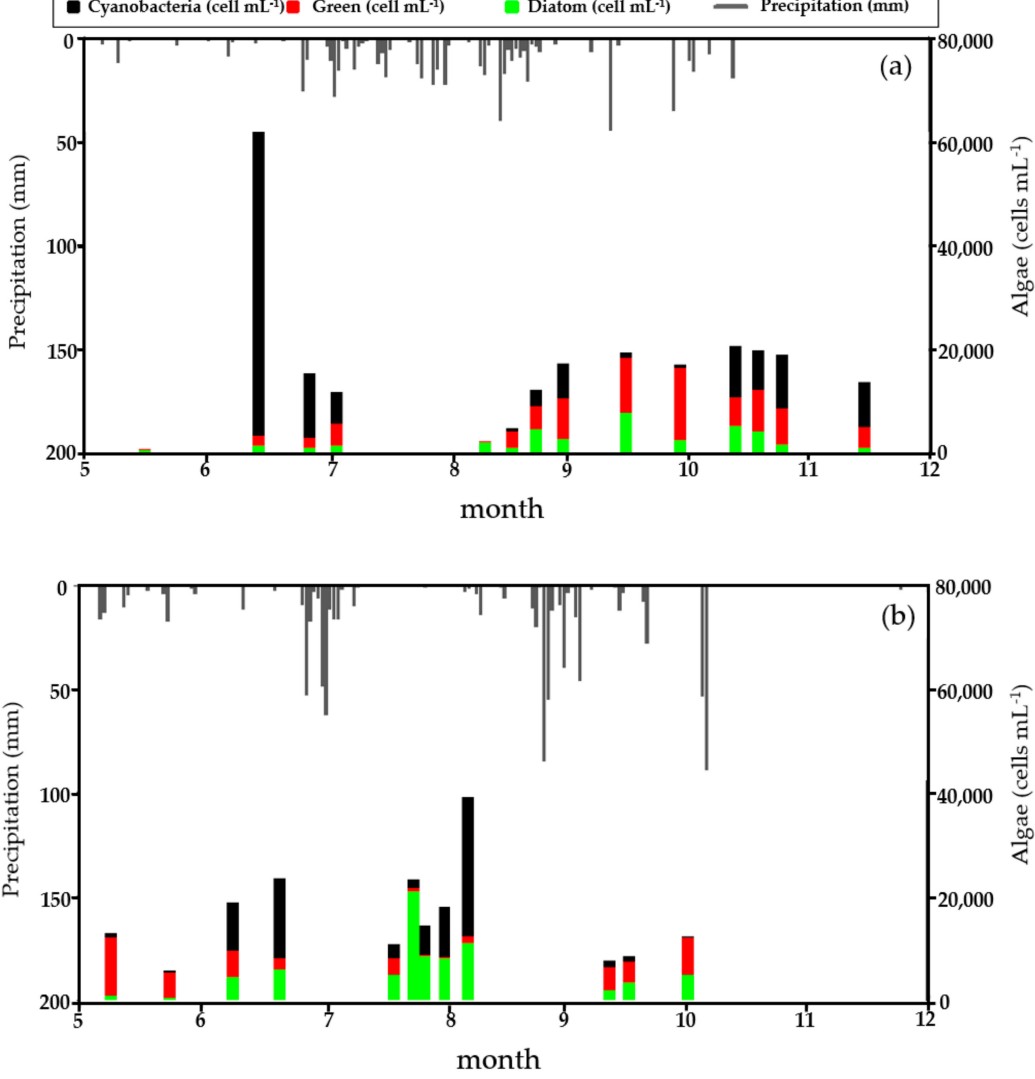

**Figure 3.** Temporal variations of precipitation and cell density (by phytoplankton group) at the Changnyeong Haman weir (CHW) in the Nakdong River from May (month 5) to December (month 12) in; (**a**) 2017 and (**b**) 2018.

*3.2. Correlation Analysis of Environmental Factors*

The Spearman correlation coefficients (*r*) among the variables are listed in Tables S1 and S2. In every weir, the cyanobacterial cell density exhibited positive correlations with Temp, ΔT, EC, and DO, and negative correlations with the flow rate and precipitation. This result coincides with the findings of previous studies [50,52], which indicate that cyanobacteria exhibit more growth than other algae when the water residence time increases, and the high temperature and stability of the water body are maintained, due to decreases in the precipitation and flow rate. The highest correlation between Cyano and Temp was obtained for GGW ($r = 0.33$, $p < 0.01$). The most significant correlations between Cyano and ΔT were obtained at GGW and HCW, with *r* values of 0.59 and 0.51, respectively ($p < 0.01$). In every weir, Cyano and Green exhibited a positive correlation. Furthermore, a positive correlation between Cyano and diatom cell density was found at GGW, whereas negative correlations were found at DSW, HCW, and CHW. This seems to have been because several algae classes were in competition during the survey period, rather than any specific class dominating [10,11].

The concentrations of Chl–a exhibited positive correlations with BOD and TOC—which indicate organic matter—in every weir. In particular, Chl–a exhibited a high positive correlation ($r = 0.56$) with BOD at GGW and HCW. This result is interpreted as indicating that the internal load of the organic matter due to algae growth in the water body increased the BOD and TOC. Furthermore, negative correlations were obtained between the Fe concentrations and the cyanobacteria, green algae, and diatom cell densities at every weir ($r = -0.34$ to $-0.07$). The photosynthesis in cyanobacteria is known to be accelerated by Fe [66]; however, no significant correlations were found. At every weir, zero or negative correlations were found between $SiO_2$ and diatom cell density, which seems to have been because the $SiO_2$ concentration was sufficiently high and did not act as a limiting factor of diatom growth. According to Wetzel [35], $SiO_2$ contributes to the seasonal transition of diatoms and other algae species, and the diatoms are weaker in terms of food competition than other algae species in a water body with a $SiO_2$ concentration of approximately 0.5 mg $L^{-1}$ or lower. In the present study, the average $SiO_2$ concentrations were very high, at 4.77 (0.04–11.38), 4.02 (0–10.18), 3.53 (0.05–9.87), and 2.77 (0–9.85) mg $L^{-1}$ at GGW, DSW, HCW, and CHW, respectively.

The correlations between the algae biomass (Chl–a) and nutrients in the study weirs are shown in Figure 4 and Table S1. The measurements showed that the TP and TN concentrations were sufficiently high to exceed the Carlson eutrophication criterion (TP = 0.03 mg $L^{-1}$, TN = 0.3 mg $L^{-1}$) in every weir. No correlation was found between Chl–a and TP, except at DSW ($r = 0.702$, $p < 0.05$). Similarly, no correlation was found between Chl–a and TN, except at DSW ($r = 0.728$, $p < 0.05$). In general, the algal biomass (Chl–a) exhibits a high correlation with TP in stagnant waters such as those of reservoirs and lakes; when the P supply is blocked by the photic zone in the reservoir surface layer, algal growth is P-limited [23,24]. However, unlike in reservoirs, nutrients are continuously supplied to the studied river from the watershed and therefore maintain a high concentration. Thus, algal growth and P concentration have a low correlation because the P-limitation on algal growth occurs only in periods when a low flow rate is sustained. It is difficult to define why algal growth was TP-limited only at DSW, however, the high TP variability (Table 1) could provide an explanation.

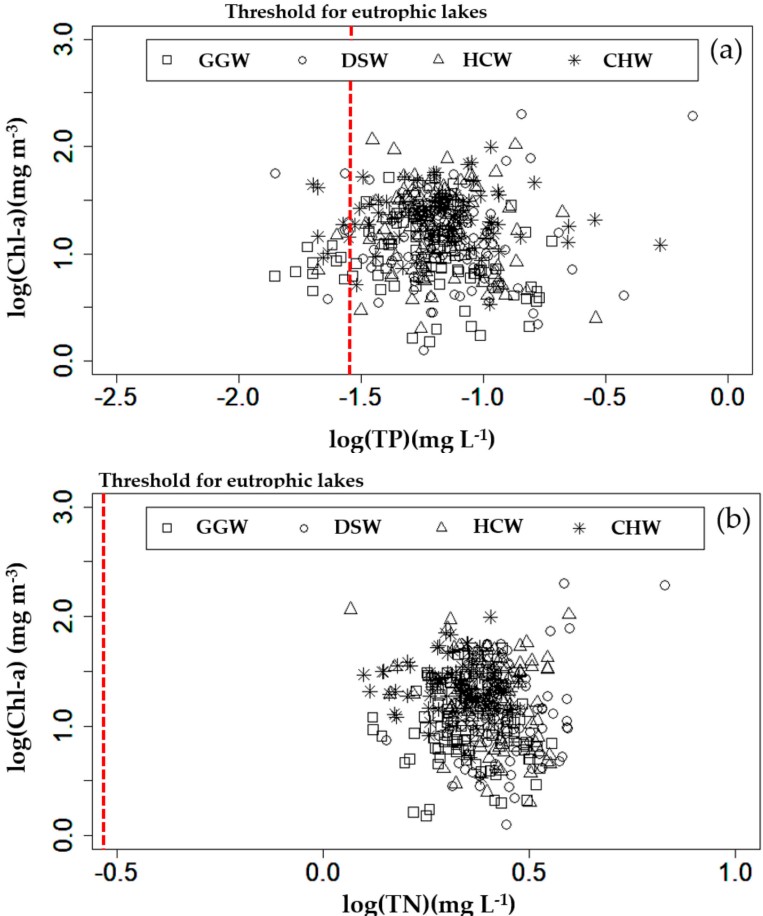

**Figure 4.** Correlations between; (**a**) Log (Chl–a) (mg m$^{-3}$) and Log (TP) (mg L$^{-1}$), and (**b**) Log(Chl–a) (mg m$^{-3}$) and Log (TN) (mg L$^{-1}$) at Gangjeong Goryeong weir (GGW), Dalseong weir (DSW), Hapcheon Changnyeong weir (HCW), and Changnyeong Haman weir (CHW).

*3.3. Selection of Important Environmental Factors Associated with Cyanobacterial Dominance*

3.3.1. Step-Wise Multiple Linear Regression (SMLR)

A parsimonious model for predicting C.dominance via the SMLR method was developed. The extraction results of the important environmental variables are listed in Table 2. The Adj.R$^2$, RMSE, Mallows' $C_P$ statistics, and AIC values of the model according to the independent variables are also presented in Table 2.

Among the 12 considered models, the best performance for GGW was obtained with EC, ΔT, NO$_3$–N, and TN as independent variables; the variability of C.dominance, (i.e., the dependent variable) was reproduced at 47.6%. Among the 12 models considered for DSW, the best performance was obtained for the model taking NO$_3$–N, EC, TN, Temp, NH$_3$–N, and Q7day as independent variables; the C.dominance variability was reproduced at 52.1%. The best performance for HCW as obtained for the model taking EC, NO$_3$–N, Temp, TN, Q7day, Chl–a, APRCP7, and Fe as independent variables; the C.dominance variability was reproduced at 72.9%. Among the 12 models considered for CHW, the best performance was obtained for the model taking NO$_3$–N, EC, APRCP7, Q7day, and TN as independent variables; the C.dominance variability was reproduced at 53.8%. The Adj.R$^2$ value of the multiple regression model selected at each weir ranged from 0.476 to 0.729, and the variability of the cyanobacterial dominance ratio was not reproduced sufficiently. These findings are interpreted as indicating that the cyanobacterial dominance environment is established through a combination of complex nonlinear relationships involving many variables, and cannot be sufficiently explained by a parametric regression model that assumes a linear combination of variables.

**Table 2.** Subset of regression variables that best-matched performance criterion in step-wise multiple linear regression (SMLR) analysis for Gangjeong Goryeong weir (GGW), Dalseong weir (DSW), Hapcheon Changnyeong weir (HCW), and Changnyeong Haman weir (CHW). Note: see the main text for definitions of abbreviated variables.

| Weir | Variables | Adj.$R^2$ | RMSE | $C_P$ | AIC |
|------|-----------|-----------|------|-------|-----|
| GGW | EC, $NO_3$–N, TN, $\Delta T$ | 0.476 | 0.195 | 4.4 | −43.9 |
| DSW | $NO_3$–N, EC, TN, Temp, $NH_3$–N, Q7day | 0.521 | 0.182 | 4.5 | −59.7 |
| HCW | EC, $NO_3$–N, Temp, TN, Q7day, Chl–a, APRCP7, Fe | 0.729 | 0.155 | 7.8 | −86.8 |
| CHW | $NO_3$–N, EC, APRCP7, Q7day, TN | 0.538 | 0.182 | 2.8 | −52.6 |

Adj.R2: Adjusted coefficient of determination; $C_P$: Mallow's $C_P$, a smaller value indicates a higher-precision model; AIC: Akaike information criterion, a smaller value indicates a higher-precision model.

### 3.3.2. Recursive Feature Elimination Based on Random Forest Model (RFE-RF)

The results of the importance evaluation of the environmental variables affecting C.dominance (%)—the dependent variable of the RFE-RF method—are outlined in Table 3. The *ntree* value of the RF model was set to an initial value of 500, as suggested by Breiman and Cutler [59], and the *mtry* value was determined via the method proposed by Liaw and Wiener [67]. The *mtry* count was set to 2–6, 2–7, 2–8, and 2–5 for GGW, DSW, HCW, and CHW, respectively, and was applied to the RF model. The simulation results show that the prediction error decreased as *mtry* increased. The lowest RMSE values at 0.079%, 0.065%, 0.063%, and 0.080% were obtained for the models with 6, 7, 8, and 5 *mtry* for GGW, DSW, HCW, and CHW, respectively. Thus, they were selected as the final parameters.

**Table 3.** Variables selected by recursive feature elimination (RFE) and order of importance for each weir; Gangjeong Goryeong weir (GGW), Dalseong weir (DSW), Hapcheon Changnyeong weir (HCW), and Changnyeong Haman weir (CHW). Note: see the main text for definitions of abbreviated variables.

| Weir | Order of variable importance | RMSE (%) |
|------|------------------------------|----------|
| GGW | EC > Temp > $\Delta T$ > Q7day > TN > $PO_4$–P > APRCP7 | 0.162 |
| DSW | EC > TOC > Temp > TP > TN > Q7day > $\Delta T$ | 0.156 |
| HCW | EC > $\Delta T$ > Temp > Q7day > APRCP7 > TOC > Fe > $PO_4$-P | 0.145 |
| CHW | EC > TN > APRCP7 > $\Delta T$ > Q7day | 0.160 |

The results show that, for GGW, the lowest RMSE of 0.162% was obtained when seven variables (EC, Temp, $\Delta T$, Q7day, TN, $PO_4$–P, and APRCP7) were used (Table 3). For DSW, the lowest RMSE of 0.156% was also obtained using seven variables (EC, TOC, Temp, TP, TN, Q7day, and $\Delta T$). For HCW, the lowest RMSE of 0.145% was obtained when eight variables (EC, $\Delta T$, Temp, Q7day, APRCP7, TOC, Fe, and $PO_4$–P) were used. For CHW, the lowest RMSE of 0.160% was obtained when five variables (EC, TN, APRCP7, $\Delta T$, and Q7day) were used.

To compare the prediction performance of the SMLR and RF models—which were both used for the variable importance evaluation—the measurements of the cyanobacteria dominance of each weir were compared with simulated values (Figure 5). The SMLR model showed a large variance with the measured values, did not accurately reflect the variability of the measured values, and had low coefficients of determination for all weirs (GGW: 0.476, DSW: 0.521, HCW: 0.729, and CHW: 0.538). In contrast, the C.dominance predicted through the parsimonious RF model accurately reflected the variability of the measured values and exhibited very high Adj.$R^2$ values compared with the SMLR model for all weirs (GGW: 0.910, DSW: 0.932, HCW: 0.949, and CHW: 0.902). This finding suggests that the RF model—which is a non-parametric method—reproduces the complex nonlinear correlations of many variables affecting the cyanobacterial dominance environment better than the SMLR model, which is a parametric method. Therefore, use of the RFE-RF method for the variable importance evaluation related to cyanobacterial dominance is considered a useful approach.

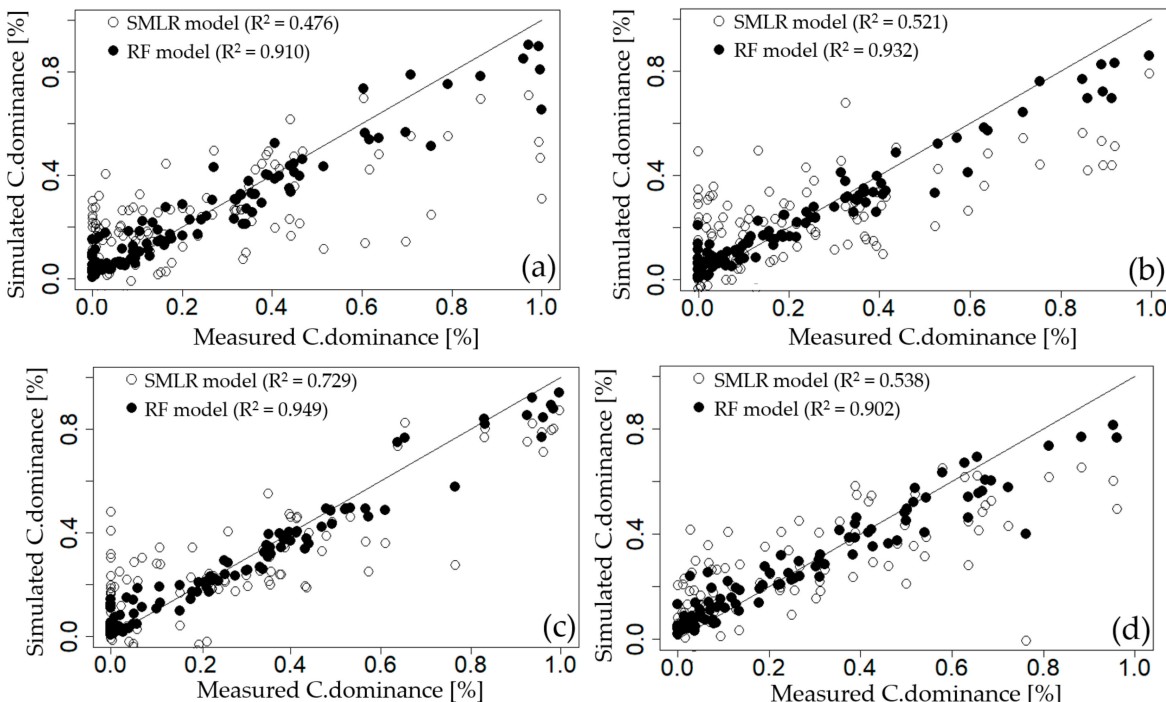

**Figure 5.** Comparison of measured cyanobacterial dominance with simulated results using step-wise multiple linear regression (SMLR) and random forest (RF) models at; (a) Gangjeong Goryeong weir (GGW), (b) Dalseong weir (DSW), (c) Hapcheon Changnyeong weir (HCW), and (d) Changnyeong Haman weir (CHW).

The partial dependence plot was used to analyze the increased conditions for cyanobacterial dominance, for important variables selected for each weir through the RFE-RF method (Figure S1). Use of a partial dependence plot is one of the methods used to identify effects of independent variables on dependent variables; in this approach, the effect of an individual independent variable on the dependent variables is visualized [59]. For GGW, the C.dominance increased with EC, ΔT, and Temp, and decreased as Q7day and TN increased. For DSW, the C.dominance increased with EC, ΔT, Temp, and TOC, and decreased with a higher Q7day. Furthermore, this property decreased when TN was lower than approximately 3 mg L$^{-1}$ and increased at higher concentrations. Furthermore, it increased in proportion to TP until the level of 0.2 mg L$^{-1}$ was reached; then, it remained constant. For HCW, the C.dominance increased with higher EC, ΔT, Temp, Fe, and APRCP7, and decreased with higher Q7day. In particular, the C.dominance increased sharply at a ΔT of 1 °C or higher. For CHW, the C.dominance increased with higher EC and when ΔT was 1 °C or higher, and decreased with higher APRCP7, Q7day, or TN concentrations.

However, the simple interpretation that the important variables for predicting cyanobacterial dominance are the cause of algal blooms is highly problematic because, although some variables may cause cyanobacterial dominance, others can appear important as a result of cyanobacterial overgrowth. For example, low PO$_4$–P in an environment with high C.dominance results from cyanobacterial overgrowth. Therefore, accurate interpretation of the data mining analysis results requires expert analysis and judgment.

### 3.4. Characterizing Environmental Conditions of High Cyanobacterial Dominance

#### 3.4.1. Decision Tree Analysis

The DT model was developed using the important variables selected by the RFE-RF method to evaluate the environmental conditions corresponding to a high cyanobacterial dominance ratio at each weir. The evaluation results obtained for each weir using the DT model are presented in Figure 6.

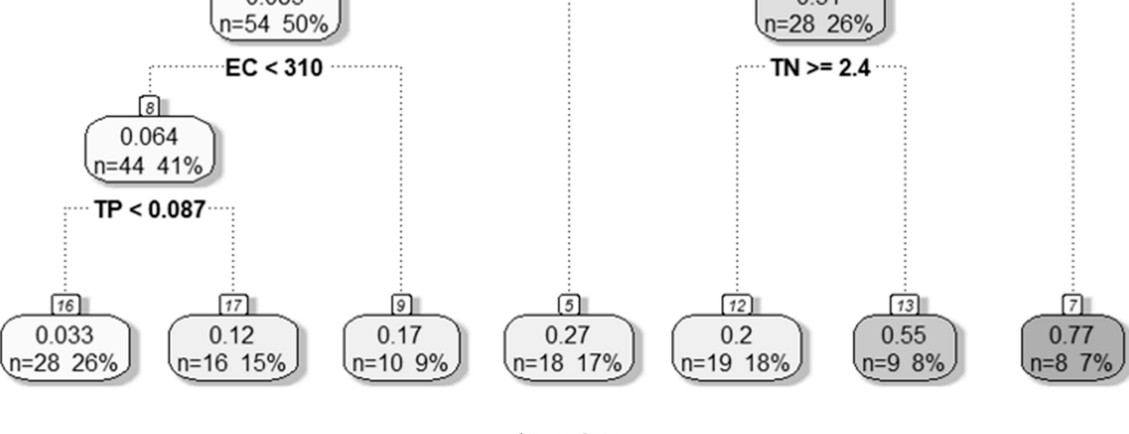

(a) GGW

(b) DSW

**Figure 6.** *Cont.*

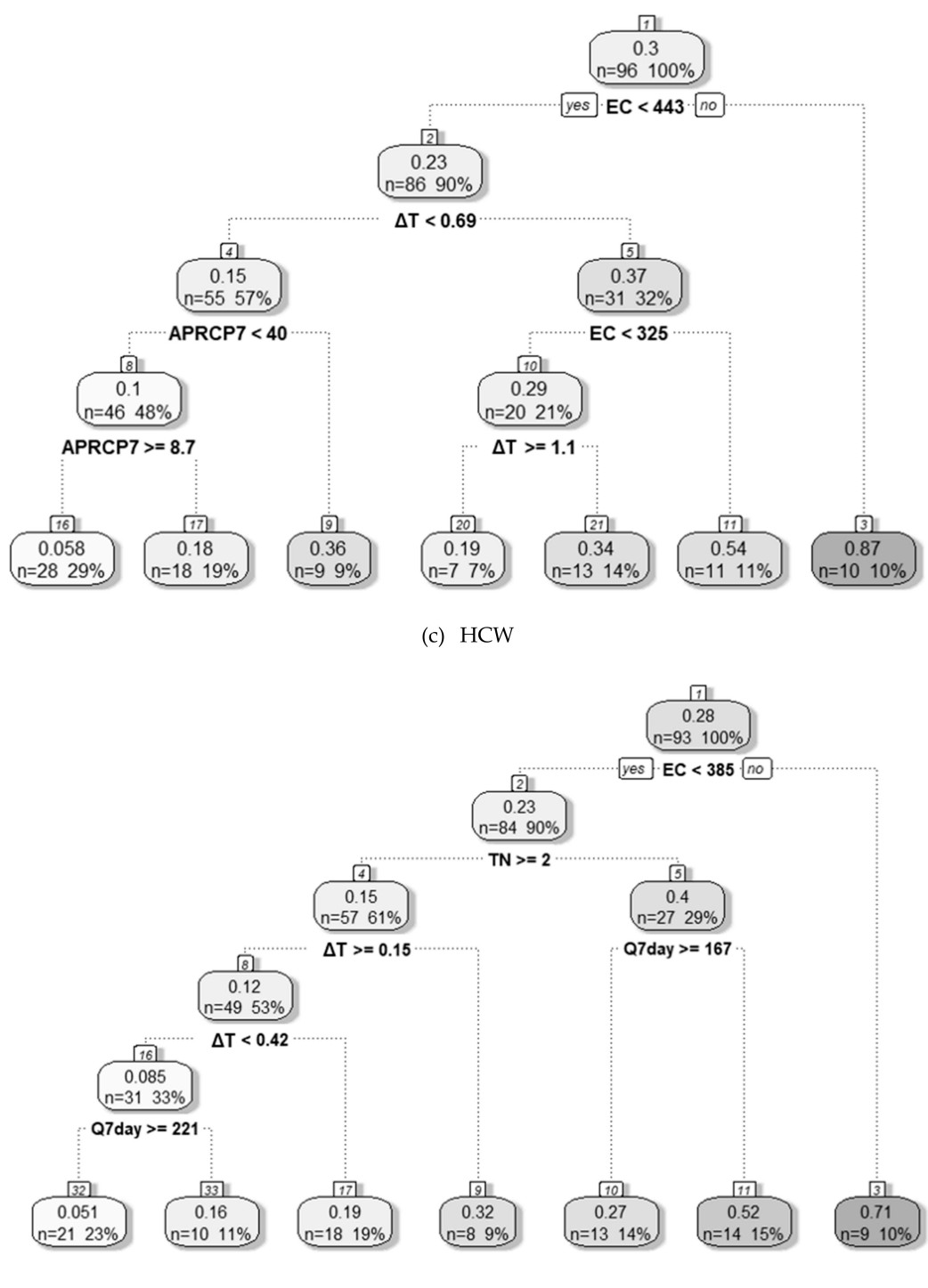

(c) HCW

(d) CHW

**Figure 6.** Evaluation of environmental conditions for high cyanobacterial dominance using a decision tree model; (**a**) Gangjeong Goryeong weir (GGW), (**b**) Dalseong weir (DSW), (**c**) Hapcheon Changnyeong weir (HCW), and (**d**) Changnyeong Haman weir (CHW). Note: see the main text for definitions of abbreviated variables.

For GGW, EC was the most important variable regarding environmental conditions influencing cyanobacterial dominance. When EC was ≥ 325 µS cm$^{-1}$, the average C.dominance was 81% in 9 out of 108 data points in total (8%). The EC was also the most important variable for DSW. In that case, when EC was ≥ 336 µS cm$^{-1}$ and TP was ≥ 0.082 mg L$^{-1}$, the average C.dominance was 77%

in 8 out of 108 data points in total (7%). When EC was $\geq$ 336 $\mu$S cm$^{-1}$, TP and TN were less than 0.082 and 2.4 mg L$^{-1}$, respectively, and the average C. dominance was 55% in 9 out of 108 data points in total (8%). For HCW, EC and $\Delta$T were important variables regarding environmental conditions for cyanobacterial dominance. The environmental conditions for a high cyanobacterial share were as follows—when EC was $\geq$ than 443 $\mu$S cm$^{-1}$, the average share was 87% in 10 out of 96 data points in total (10%). When EC was < 443 $\mu$S cm$^{-1}$ and $\geq$ 325 $\mu$S cm$^{-1}$, and when $\Delta$T was $\geq$ 0.7 °C, the average share was 54% in 11 out of 96 data points in total (11%). For CHW, EC, TN, and Q7day were found to be important environmental variables for with the growth of a high cyanobacterial population. When EC was $\geq$ 385 $\mu$S cm$^{-1}$ at CHW, the average share was 71% in 9 out of 93 data points in total (10%). When EC was < 385 $\mu$S cm$^{-1}$, TN was < 2 mg L$^{-1}$, and Q7day was < 167 m$^3$ s$^{-1}$, the average share was 52% in 14 out of 93 data points in total (15%).

Importantly, the greater influence of EC compared to other environmental conditions in facilitating high cyanobacterial population growth is because the EC is susceptible to flow rate variations, and not because it promotes cyanobacterial growth (Figure 7). As shown in Figure 7, EC tended to increase as the river flow decreased at each weir. Thus, EC can be considered an indirect index indicating a continuous low flow condition in the river. In other words, when drought persists and the natural runoff quantity due to rainfall-runoff is insufficient, EC increases. This is because the river water quality is greatly affected by the groundwater and effluent from large sewage treatment plants. Furthermore, EC may increase due to anaerobicization of the water-sedimentary interface due to thermal stratification and the subsequent release of ionic materials [68,69].

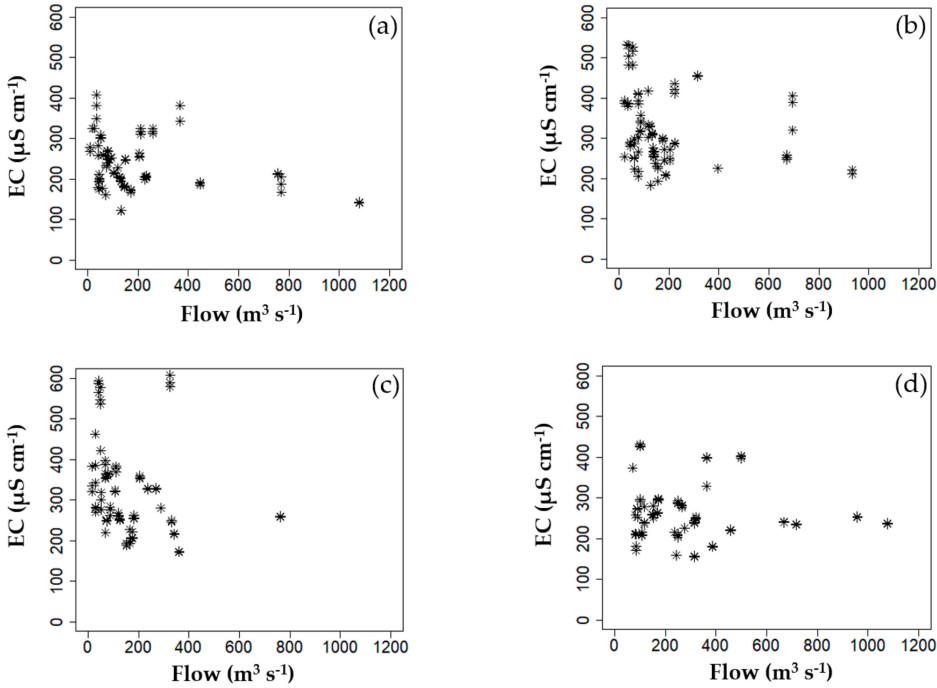

**Figure 7.** Correlation between electrical conductivity (EC) ($\mu$S cm$^{-1}$) and flow rate (m$^3$ s$^{-1}$) at; (**a**) Gangjeong Goryeong weir (GGW), (**b**) Dalseong weir (DSW), (**c**) Hapcheon Changnyeong weir (HCW), and (**d**) Changnyeong Haman weir (CHW).

### 3.4.2. Principal Component Analysis

PCA was performed for the cluster analysis of the variables that had large correlations with a cyanobacterial dominance environment, and to verify the environmental characteristics corresponding to a cyanobacterial cell density of 10,000 cells mL$^{-1}$ or higher. The principal axes with eigenvalues of 1.0 or higher in the PCA results were extracted as the main components. Hence, five, two, five, and five principal components with eigenvalues of 1.0 or higher were selected for GGW, DSW, HCW,

and CHW, respectively. The analysis showed that for GGW, the first, second, third, fourth, and fifth principal components contributed 31.9%, 18.8%, 9.6%, 8.2%, and 6.7%, respectively, to the total water quality change. For DSW, the first and second principal components contributed 44.3% and 21.7%, respectively, and for HCW, the first, second, third, fourth, and fifth principal components contributed 21.9%, 20.1%, 15.8%, 9.3%, and 7.4%, respectively, to the total water quality change. For CHW, the first, second, third, fourth, and fifth principal components contributed 25.2%, 18.9%, 12.6%, 8.2%, and 6.1%, respectively, to the total water quality change.

For each weir, the PCA score of the measurement data and the loading (contribution) vector of each variable are represented in bi-plots (Figure 8). For data grouping, the harmful cyanobacterial cell density levels (harmful algal bloom [HAB] level) are shown together, divided into normal (lower than 1,000 cells mL$^{-1}$), warning (1,000–10,000 cells mL$^{-1}$), and alarm (higher than 10,000 cells mL$^{-1}$) levels.

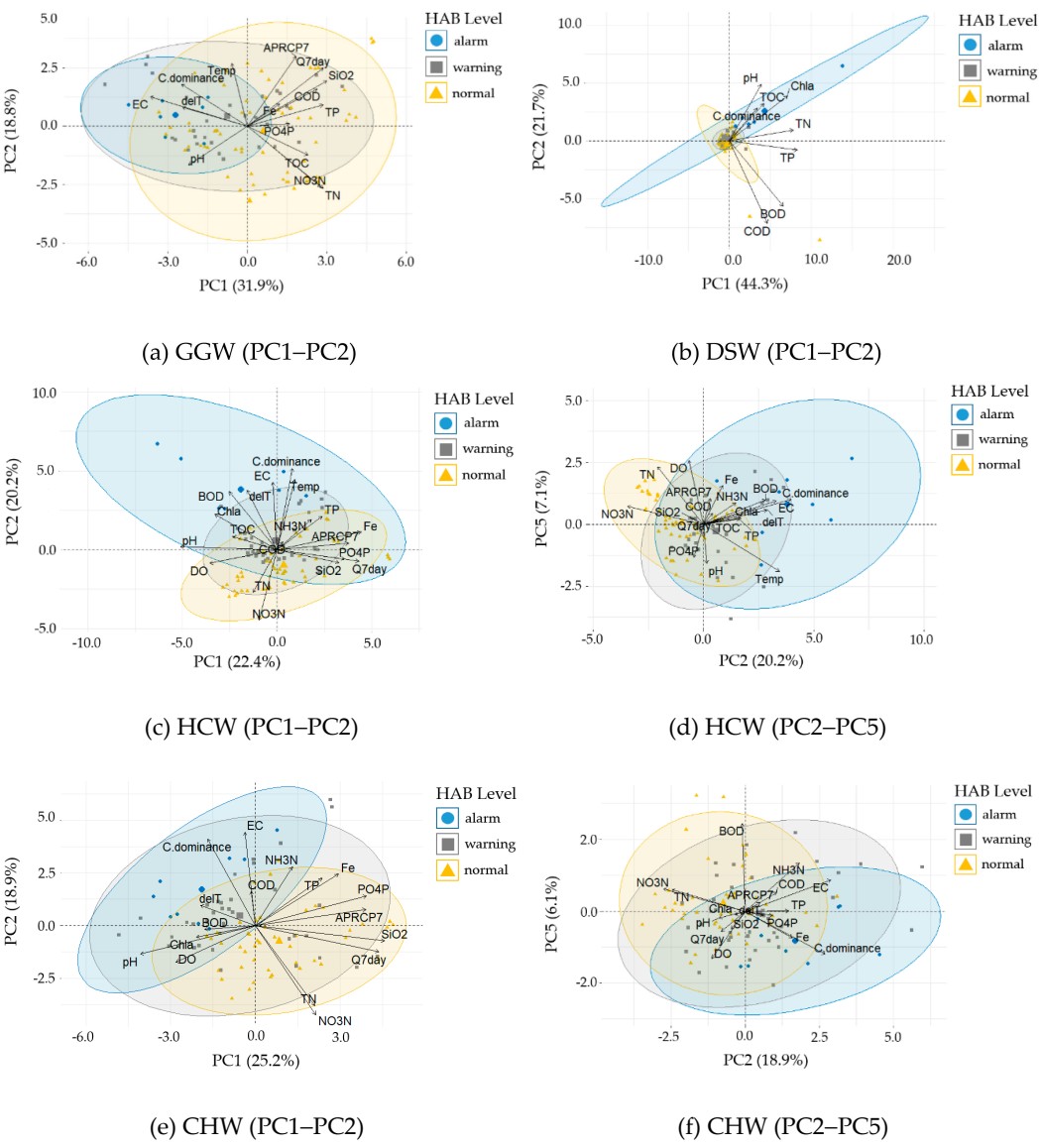

**Figure 8.** Principle component analysis (PCA) bi-plots grouped by the harmful algal bloom (HAB) level. The larger symbols indicate the group mean point for each HAB level. (**a**) Gangjeong Goryeong weir (GGW; PC1-PC2), (**b**) Dalseong weir (DSW; PC1-PC2), (**c**) Hapcheon Changnyeong weir (HCW; PC1-PC2), (**d**) HCW (PC2-PC5), (**e**) Changnyeong Haman weir (CHW: PC1-PC2), and (**f**) CHW (PC2-PC5).

In the GGW (Table 4), EC, C.dominance, ΔT, and pH have the greatest effect on the negative direction of the first principal component axes (PC1), while $SiO_2$, Q7day, TN, TP, $NO_3$–N, COD, and TOC have the greatest effect on the positive direction of PC1. As apparent from the bi-plot for PC1 and PC2, APRCP7, Q7day, $SiO_2$, and COD form the same cluster in the opposite direction of pH (Figure 8a). Furthermore, the APRC7 and Q7day values for the summer can be divided into large and small values. The cyanobacteria tended to dominate when the precipitation and flow were low. The C.dominance vector has the same direction as EC, ΔT, and Temp, thus explaining the high EC, strong thermal stratification, and high cyanobacterial share at high temperatures (Figure 8a). The data corresponding to a cyanobacterial cell density higher than 10,000 cells $mL^{-1}$ mainly appear under high-EC conditions, strong thermal stratification, and a high cyanobacterial share. The results are in a good agreement with the findings in the lowland rivers of south-eastern Australia [10,11,13], where a persistent low flow and the formation of thermal stratification caused the proliferation of cyanobacteria.

In the DSW (Table 4), TP, TN, Chl–a, BOD, COD, and pH have the greatest effect on the positive direction of PC1, while COD and BOD have the greatest effect on the negative direction of PC2. C.dominance has the largest effect on PC1 and PC2. In the bi-plot for PC1 and PC2, the C.dominance loading vector forms a cluster in the same direction as pH, TOC, and Chl–a (Figure 8b). Furthermore, the data corresponding to a cyanobacterial cell density higher than 10,000 cells $mL^{-1}$ appear under the conditions of high TOC, pH, Chl–a, and cyanobacterial share.

**Table 4.** Contributions of variables to principal components; Gangjeong Goryeong weir (GGW), Dalseong weir (DSW), Hapcheon Changnyeong weir (HCW), and Changnyeong Haman weir (CHW). Note: see the main text for definitions of abbreviated variables.

| Variable | GGW | | DSW | | HCW | | CHW | |
|---|---|---|---|---|---|---|---|---|
| | Component | | Component | | Component | | Component | |
| | 1 | 2 | 1 | 2 | 1 | 2 | 1 | 2 |
| C.dominance | 6.75 | 6.00 | 3.99 | 5.61 | 0.42 | 19.26 | 2.10 | 16.77 |
| APRCP7 | 3.69 | 16.42 | - | - | 9.29 | 0.12 | 11.40 | 0.58 |
| Q7day | 8.97 | 11.53 | - | - | 12.24 | 0.39 | 14.26 | 1.54 |
| Temp | 0.39 | 13.41 | - | - | 0.57 | 14.55 | - | - |
| ΔT | 6.56 | 1.20 | - | - | 1.65 | 10.31 | 2.79 | 0.91 |
| DO | - | - | - | - | 8.21 | 0.54 | 5.87 | 2.98 |
| pH | 5.40 | 5.07 | 5.15 | 16.98 | 17.04 | 0.03 | 12.23 | 1.81 |
| EC | 14.66 | 2.92 | - | - | 0.03 | 13.27 | 0.10 | 19.43 |
| BOD | - | - | 15.01 | 22.48 | 4.21 | 9.75 | 2.38 | 0.03 |
| COD | 7.59 | 4.76 | 7.51 | 35.81 | 0.05 | 0.19 | 0.01 | 2.60 |
| TOC | 5.65 | 2.87 | 6.19 | 7.36 | 3.62 | 0.53 | - | - |
| TP | 8.90 | 1.54 | 23.80 | 0.46 | 3.70 | 3.25 | 4.15 | 5.01 |
| $PO_4$–P | 2.76 | 0.02 | - | - | 8.25 | 0.24 | 11.49 | 1.97 |
| TN | 8.93 | 13.07 | 20.99 | 0.62 | 1.09 | 5.35 | 3.32 | 14.21 |
| $NH_3$–N | - | - | - | - | 2.15 | 1.28 | 1.28 | 7.73 |
| $NO_3$–N | 8.36 | 12.49 | - | - | 0.60 | 14.54 | 3.42 | 17.72 |
| Fe | 1.60 | 1.68 | - | - | 12.88 | 0.92 | 6.42 | 5.89 |
| $SiO_2$ | 9.79 | 7.01 | - | - | 6.98 | 0.47 | 15.55 | 0.52 |
| Chl–a | - | - | 17.36 | 10.68 | 7.01 | 3.63 | 3.23 | 0.29 |

Note: - missing values are variables that were not included in the PCA because the KMO test value < 0.5.

In the HCW (Table 4), Fe, Q7day, APRCP7, $PO_4$–P, and $SiO_2$ have the greatest effect on the positive direction of PC1, while pH, DO, and Chl–a have the greatest effect on the negative direction. C.dominance has the largest effect on the PC2 axis, whereas APRC7 and Q7day have the largest effects on the PC1 and PC3 axes. The PC1 axis corresponds to conditions of large precipitation and flow, which mean inward flow of nonpoint pollutant sources; this explains the high TP, $PO_4$–P, Fe, and $SiO_2$ concentrations. From the examination of the bi-plot on the PC1–PC2 plane (Figure 8c), the cyanobacterial share loading vector of HCW has the same direction as EC, Temp, and ΔT, and the opposite direction to the $NO_3$–N and TN vectors. This explains the high C.dominance under the conditions of high EC, water temperature, and stratification strength. On the PC2–PC5 plane, the

C.dominance vector has the same cluster direction as EC, ΔT, BOD, Chl–a, Temp, and $NH_3$–N. The alarm data (corresponding to a cyanobacteria cell density higher than 10,000 cells $mL^{-1}$) appeared when these variables were high.

In the CHW (Table 4), $SiO_2$, Q7day, APRCP7, $PO_4$–P, and Fe have the greatest effect on the positive direction of PC1, while pH and DO have the greatest effect on the negative direction of PC1. On the PC2 axes, EC, C.dominance, Fe, TP, and $NH_3$–N have the greatest effect on the positive direction, while $NO_3$–N and TN have the greatest effect on the negative direction. In the bi-plots (Figure 8e,f), the C.dominance has the largest effect on the PC2 axis, whereas APRC7 and Q7day have the largest effects on the PC1 and PC4 axes. On the PC1–PC2 plane, the APRC7 and Q7day vectors form clusters in the same direction as $PO_4$–P, $SiO_2$, TP, and Fe, and in the opposite direction to pH, Chl–a, C.dominance, ΔT, BOD, and DO (Figure 8e). In the CHW bi-plots, the alarm data corresponding to a cyanobacterial cell density higher than 10,000 cells $mL^{-1}$ appeared for conditions where the cyanobacterial share, EC, ΔT, and Fe were high (Figure 8e,f).

*3.5. Integrated Analysis of Environmental and Control Variables for Cyanobacterial Dominance at Each Weir*

To present a general evaluation of the cyanobacterial dominance environment and the effective algal bloom control variables for each weir, the integrated results of the above analyses are listed in Table 5. For the integrated evaluation, the variable importance rank selected by the correlation analysis (*r* >|0.5|), the RFE-RF, the 50% or higher cyanobacterial share condition of the DT model, and the PCA results were used. The SMLR model results were excluded from the integrated analysis because the SMLR prediction performance for the cyanobacterial dominance ratio was lower than that of the RFE-RF.

The integrated analysis results show that the variables that had high correlations with the cyanobacterial dominance environment at GGW were EC and ΔT, which correspond to an environment with consistently low flow and a stable water body. This result reflects the favorable growth characteristics of cyanobacteria in a stable water body with a higher temperature compared with diatoms and green algae [61,62]. The EC increase in the river tends to increase as precipitation and natural flow decrease. Thus, it is appropriate to consider EC as a factor for indicating continuous drought and low flow conditions. Contrary to expectation, EC had a greater effect than Q7day, which is a direct flow index. This is because Q7day indicates the flow conditions over a short period of seven days, during which the flow is affected by the discharge rate of the upstream dam, whereas EC indirectly indicates the long-term reduction of the rainfall-runoff and natural flow. Furthermore, EC may be increased by anaerobicization of the water-sedimentary layer interface due to thermal stratification and the subsequent release of ionic materials [70,71]. Therefore, the factors having the largest effects on the algal bloom due to cyanobacteria at GGW are considered to be physical factors such as the natural flow reduction, water temperature increase, and thermal stratification strengthening.

The cyanobacterial dominance at DSW was associated with EC, TOC, Temp, ΔT, TP, and TN. The C.dominance was high when the EC and TP concentrations were high. There are many industrial complexes and urban areas around the Geumho River, which flows into DSW. The treated water released from the sewage treatment plants contains high nutrients and accounts for more than 40% of the water quantity in the downstream Geumho River, which flows into the mainstream Nakdong River [54]. The nutrients that flow from the Geumho River to DSW can accelerate the growth of algae flowing down from upstream [72]. Therefore, DSW exhibited a higher degree of organic-matter and nutrient pollution than the other weirs (Table 1), and the TN and TP concentrations exhibited a high correlation with the Chl–a concentration (Table S1). This result supports the action of nutrients as a limiting factor for algal growth at DSW. Therefore, to suppress algal blooms due to overgrowth of cyanobacteria, the appropriate flow rate must be maintained and the TP load must be reduced.

**Table 5.** Integration of statistical analysis and data mining results for comprehensive interpretation; Gangjeong Goryeong weir (GGW), Dalseong weir (DSW), Hapcheon Changnyeong weir (HCW), and Changnyeong Haman weir (CHW). Note: see the main text for definitions of abbreviated variables.

| Weir | Correlation Analysis | | | | Recursive Feature Elimination | | | | | Decision Tree | | PCA | |
| | (r > \|0.5\|) | | | | (Variable importance rank) | | | | | (C.dominance > 50% conditions) | | (Clustering) | |
| | 1st | 2nd | 3rd | 4th | 1st | 2nd | 3rd | 4th | 5th | 1st | 2nd | Positive | Negative |
|---|---|---|---|---|---|---|---|---|---|---|---|---|---|
| GGW | Chl–a | TOC | $\Delta$T | - | EC | $\Delta$T | TN | Temp | Q7day | EC $\geq$ 325 $\mu$S cm$^{-1}$ | - | EC, $\Delta$T, Temp | TOC NO$_3$–N TN |
| DSW | Chl–a | SS | TN | TP | EC | TOC | Temp | $\Delta$T | TP | EC $\geq$ 336 $\mu$S cm$^{-1}$ TP $\geq$ 0.082 mg L$^{-1}$ | EC $\geq$ 336 $\mu$S cm$^{-1}$ TP < 0.082 mg L$^{-1}$ TN < 2.4 mg L$^{-1}$ | pH, TOC Chl–a | - |
| HCW | $\Delta$T BOD | - | - | - | EC | $\Delta$T | Temp | Q7day | Fe | EC $\geq$ 443 $\mu$S cm$^{-1}$ | 325 $\mu$S cm$^{-1}$ $\leq$ EC < 443 $\mu$S cm$^{-1}$ $\Delta$T $\geq$ 0.7 °C | EC Temp $\Delta$T BOD NH$_3$–N Chl-a | NO$_3$–N TN |
| CHW | - | - | - | - | EC | $\Delta$T | TN | Q7day | APRCP7 | EC $\geq$ 385 $\mu$S cm$^{-1}$ | EC < 385 $\mu$S cm$^{-1}$ TN < 2 mg L$^{-1}$ Q7day < 167 m$^3$ s$^{-1}$ | EC $\Delta$T Fe | NO$_3$–N TN |

For HCW, the variables with high correlation with the cyanobacterial dominance environment were EC, ΔT, Temp, Q7day, and Fe. The C.dominance was high in an environment with a small flow rate and a stable water body. This is similar to the GGW result and suggests that physical factors have greater effects on cyanobacterial dominance. However, the cumulative effect of the river flow rate and thermal stratification on the cyanobacterial dominance is more important than the temporary effects [10,13]. Therefore, the continuous cumulative stratification strength based on the high-frequency sensor data should be used as an input variable.

The variables that had high correlations with the cyanobacterial dominance environment at CHW were EC, ΔT, TN, Q7day, and APRCP7. The C.dominance was high in an environment with high EC and low Q7day. At DSW and CHW, the TN concentration tended to be low under the cyanobacterial dominance conditions. This is because a change in TN concentration in a river is mostly determined by a change in the $NO_3$–N concentration, and the $NO_3$–N concentration is high in spring and autumn, when the base outflow is more dominant than in summer (when rainfall is concentrated). However, care should be taken in interpreting this result, because $NO_3$–N is depleted in the algal growth stage and has a low concentration when cyanobacterial cell density is high.

To combine the findings regarding the cyanobacterial dominance environments of all weirs, the cyanobacteria were dominant at CHW, DSW, GGW, and HCW when the natural flow of the river was low, the water temperature of the water body was high, and strong thermal stratification persisted. For DSW, a high TP concentration was also among the important cyanobacterial dominance causation factors.

## 4. Conclusions

In this study, the weather, flow, water quality, and algae measurement data collected at four weirs (GGW, DSW, HCW, and CHW) installed on the Nakdong River from May 16, 2017, to November 23, 2018, were analyzed using data mining techniques. The important variables related to cyanobacterial dominance were extracted for each weir and the causes of algal bloom were analyzed in an integrated manner.

The results show that the environmental factors of cyanobacterial dominance differed for each weir. The important environmental variables selected from the RFE-RF analysis were, in descending order of importance, EC, ΔT, TN, Temp, and Q7day for GGW; EC, TOC, Temp, ΔT, and TP for DSW; EC, ΔT, Temp, Q7day, and Fe for HCW; and EC, ΔT, TN, Q7day, and APRCP7 for CHW. The SMLR method applied for the variable importance evaluation had a lower prediction performance than the RFE-RF method. This is because the cyanobacterial dominance environment is determined according to the complex nonlinear relationships involving many variables and is difficult to explain through the use of a regression model that assumes a linear combination of variables.

When the environmental conditions of cyanobacterial dominance were analyzed using the DT method, this dominance was found to occur at GGW, DSW, CHW, and HCW when the natural flow of the river was low (indicated by a high EC), the water temperature was high, and strong thermal stratification persisted. For DSW, a high TP concentration was also among the important conditions. Similarly, the PCA results indicated high cyanobacterial dominance at GGW, CHW, and HCW when the rainfall-runoff was low in summer, the water temperature was high, and the thermal stratification was strong. Under these conditions, the measurement data corresponding to a cyanobacterial cell density exceeding 10,000 cells mL$^{-1}$ formed a cluster. Therefore, securing an appropriate flow rate and removing thermal stratification are critical measures for controlling algal blooms at every weir examined in this study, and various efforts including TP load reduction are required at DSW.

The EC exhibited a close positive correlation with C.dominance at every weir. The EC in a river tends to increase when the groundwater discharge and effluent from sewage treatment plants have large effects on the river flow rate because of a reduction in the natural rainfall-runoff. Therefore, EC can be regarded as an indirect index indicating continuous drought in the watershed. As EC can be measured with sensors, it can be used as the most important variable for prediction of algal blooms,

providing high-frequency measurement data that can be secured in real time together with water temperature data.

In contrast to the fact that the TP concentration—a chemical factor—is an important limiting factor for algal growth in most reservoirs in South Korea, the correlation between TP and Chl–a was very low at the weirs considered in this study, except for DSW ($p < 0.05$). This is because a high TP concentration was maintained in the study river (Nakdong River), at a level that does not limit the algal growth (>0.03 mg L$^{-1}$). The TP concentration only limits the algal growth during certain periods when the algal biomass increases sharply.

The sequential data analysis method proposed in this study is applicable to other rivers and lakes with serious algal blooms due to cyanobacterial dominance, and can be employed as a useful tool for analyzing the causes of algal blooms and establishing optimal control alternatives considering the weather, hydrology, and water quality characteristics of the target water body.

**Supplementary Materials:** The following are available online at http://www.mdpi.com/2073-4441/11/6/1163/s1, Figure S1—Partial dependence plots of random forest (RF) models, showing marginal effects of single variables on cyanobacterial dominance (C.dominance) at; (a) Gangjeong Goryeong weir (GGW), (b) Dalseong weir (DSW), (c) Hapcheon Changnyeong weir (HCW), and (d) Changnyeong Haman weir (CHW). Note: see the main document for definitions of variables. Table S1—Bi-variable correlation analysis between parameters relevant to cyanobacterial estimation at the Gangjeong Goryeong weir (GGW; right, gray) and the Dalseong weir (DSW; left, white). Note: see the main document for definitions. Table S2—Bi-variable correlation analysis between parameters relevant to cyanobacterial estimation at Hapcheon Changnyeong weir (HCW) (right, gray), and Changnyeong Haman weir (CHW) (left, white). Note: see the main document for definitions. Table S3—Correlation analysis between nutrients (total phosphorous, TP; and total nitrogen, TN) and chlorophyll–a (Chl–a) at the Gangjeong Goryeong weir (GGW), the Dalseong weir (DSW), the Hapcheon Changnyeong weir (HCW), and the Changnyeong Haman weir (CHW).

**Author Contributions:** Conceptualization, S.C.; Data curation, S.C.; Formal Analysis, S.K.; Funding Acquisition, S.C.; Investigation, H.P., Y.C., H.L.; Project Administration, H.L.; Supervision, S.C.; Visualization, S.K., H.P.; Writing, Original Draft Preparation, S.K.; Writing, Review and Editing, S.C., Y.C., H.L.

**Funding:** This work is supported by K-water (C5201902341), partly by the Korea Environmental Industry & Technology Institute (KEITI) grant funded by the Ministry of Environment (RE201901083), and partly by the Basic Science Research Program of the National Research Foundation of Korea (NRF), which is funded by the Ministry of Education (Grant number 2016R1D1A3B03932308).

**Conflicts of Interest:** The authors declare no conflict of interest.

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
