# Peer review of "Analysis of Environmental Factors Associated with Cyanobacterial Dominance after River Weir Installation"

_water, doi:10.3390/w11061163_

Round 1

Reviewer 1 Report

The article is interesting from the point of view of both basic and applied research. It includes a comprehensive and innovative approach to the problem of algae blooms near weirs. Various literature has been used for this purpose (scientific articles, government publications, thematic books), on the basis of which an existing environmental problem, i.e. cyanobacterial blooms after river weir installation, has been presented. To realize the assumed topic, extensive hydrological and physicochemical analyzes were carried out using specialized equipment. Based on them, a statistical analysis was performed, i.e. correlation analysis, recursive feature elimination, decision tree and PCA. Based on this analysis, the authors drew in a detailed way conclusions on how and on what scale the effects of cyanobacteria are visible in the context of their impact on various environmental variables. The presented contents are clear, well described and sufficiently explained, understandable even for non-specialists. Their quality is high. The text itself is interestingly presented, has a logically arranged layout that does not raise any objections. The scientific value is high, as is the importance of the content presented for the development of not only science, but also industry (the analysis can be used by people associated with hydrological research or water quality, as well as by specialized companies involved in the development of technological solutions for water purification at example for the purpose of obtaining water suitable for consumption, similarly this analysis will be a valuable hint for people involved in the maintenance and management of water facilities on watercourses). The introduction contains a comprehensive overview of the literature introducing the issue and explaining all the intricacies of the topic, the admission is transparent and clear. The designed experience is correct in terms of content - it is recommended to continue the research, it is worth focusing in particular on the parameters that significantly contribute to algal blooms; one can also think about increasing the frequency of research and the density of the research cross-section network - for example above and below weirs, at various depths, at different distances from these hydrotechnical constructions. I am just asking for an explanation of why a distance of 500 m from weirs has been accepted in the study. The methods used were presented in a sufficient way, they were provided with tables and charts, which well illustrate their use and obtained results. The results themselves are described equally well and they do not raise objections - they adequately show the dependences described in the article and organize them, proving the validity of the literature referred to, but also constituting a starting point for further scientific research. The conclusions result from the quoted results, they are clear and their description is sufficient to exhaust the topic. The language used is rich, suitable for the presented content, does not contain any grammatical or editorial errors. The language level of the article is high, yet it remains understandable for a wide audience. There was no conflict of interest, plagiarism or other ethical premises that would prevent publication of the article. I recommend publication of the article in an unchanged form, without any corrections.

Author Response

Thank you very much for your careful review and suggestions that have helped us improve the quality of our manuscript. Reviewers’ comments/suggestions are in bold text. Author responses are in plain text. All revisions are highlighted in the revised manuscript

Response to Reviewer #1

It is recommended to continue the research, it is worth focusing in particular on the parameters that significantly contribute to algal blooms; one can also think about increasing the frequency of research and the density of the research cross-section network - for example above and below weirs, at various depths, at different distances from these hydrotechnical constructions. I am just asking for an explanation of why a distance of 500 m from weirs has been accepted in the study

Authors’ response:

First of all, we appreciate this feedback. The samples for analysis were collected at a point 500 m upstream of the four study weirs because they are in the same position as the regular water quality monitoring stations operated by the Ministry of Environment of Korea. This information has now been added to the manuscript to provide clarity [p. 4 L156-157].

Reviewer 2 Report

An excellent piece of work.Exceptionally thorough in a complex area of observation.very detailed analysis.

Author Response

Thank you very much for your careful review and suggestions that have helped us improve the quality of our manuscript. Reviewers’ comments/suggestions are in bold text. Author responses are in plain text. All revisions are highlighted in the revised manuscript.

Response to Reviewer #2

An excellent piece of work. Exceptionally thorough in a complex area of observation, very detailed analysis.

Authors’ response:

We would like to express appreciation for this positive feedback on this study.

Reviewer 3 Report

Line 15.  There were no field experiments.  Suggest revising to:  This study analyzed environmental factors associated with cyanobacterial dominance at four weirs installed in Nakdong River through analysis of field collected data using parametric and non-parametric data mining methods.

Line 131 Delete “experimental”

Line 138-140 Delete reference to experiments.  Maybe replace “experiments on” with “measurements of”.

I don’t think the complete correlation matrices in Tables 2 and 3 are necessary.  These tables also have some overlap with the information in Fig. 4 and Table 4.  At most the correlations with the algal taxonomic groups might be shown, which would be the last 3 columns or rows of Tables 2 and 3.

Lines 313-316.  Correlations of chl-a with BOD and TOC not of much interest – recommend deleting this discussion.

Line 329.  Delete “experiment”.

Recommend deleting Table 4.  These data are also shown in Figure 4, and the few significant correlations are also given in the text.

Table 5.  I don’t think it is necessary to show the regression statistics for all 48 SMLR models tested.  It is sufficient to show the best model for each site.

Some of the discussion of the PCA analysis doesn’t make sense to me, regarding which factors have the greatest effect.  For example, for GCW lines 510-511 state that C.dominance and Q7day have the greatest effect on the 1st and 2nd axes.  However, Table 7 appear to show higher contributions for other factors, including EC and SiO2 on axis 1, and Temp, TN, and NO3N on axis 2.  Similarly for DSW, line 520 says C.dominance has greatest effect on the 1st and 2nd axes.  However, in Table 7 several other factors have larger contributions on both axes.

Figure 9, a few of the points are larger.  Is there any meaning to that?  They do not appear to be samples with particularly high cyanobacteria, since some of them have a normal HAB level.  Also, the shaded areas appear to be intended to include all the samples with a particular HAB level – were they just drawn to illustrate that or was some statistical procedure used to develop them?

Inconsistency – Line 640 argues that TP is high enough that it doesn’t limit algal growth.  However, DSW, the one site that TP appeared to be an important factor, had the highest mean TP concentration.  I note that Table 1 indicates DSW also has the largest standard deviation for TP, so maybe that higher variability means that site oscillates between TP limitation and excess, while the other sites more consistently have excess TP?

Author Response

Thank you very much for your careful review and suggestions that have helped us improve the quality of our manuscript. Reviewers’ comments/suggestions are in bold text. Author responses are in plain text. All revisions are highlighted in the revised manuscript.

Response to Reviewer #3

There were no field experiments. Suggest revising to: This study analyzed environmental factors associated with cyanobacterial dominance at four weirs installed in Nakdong River through analysis of field collected data using parametric and non-parametric data mining methods

Authors’ response: [p.1 L13, p.3 L121, p.3. L127, p.5. L183-184, p.10. L327]

The reviewer makes a valid point, and we agree. Hence, the term “experiments” used in the manuscript has been amended to “measurements” in all instances. To clarify, in this study, all water quality and algae data, except for the meteorological and flow data, used for statistical analysis were collected by the authors through field monitoring and laboratory measurements.

Delete “experimental”

Authors’ response: [p.3 L121]

The term “experimental” has all been corrected to “laboratory measurements” in all instances.

Delete reference to experiments. Maybe replace “experiments on” with “measurements of”.

Authors’ response: [p.3 L 127]

We have replaced “experiments on” with “measurements of”.

I don’t think the complete correlation matrices in Tables 2 and 3 are necessary. These tables also have some overlap with the information in Fig. 4 and Table 4. At most the correlations with the algal taxonomic groups might be shown, which would be the last 3 columns or rows of Tables 2 and 3.

Authors’ response: [supplementary material]

Tables 2 and 3 have been removed from the manuscript, instead we have now included them in the supplementary material as Table S1 and Table S2.

Correlations of Chl-a with BOD and TOC not of much interest– recommend deleting this discussion

Authors’ response: [p.9 L 312-315]

While we appreciate and respect the Reviewer’s comment, we have retained the information regarding the correlations of Chl-a with BOD and TOC because we firmly believe that these correlations are highly relevant for determining the effect of weir installation on the increase in internal organic load by primary productivity in the river. However, should the Editor consider this information to not be of much interest after consideration of our rebuttal, we will be more than happy to reconsider this decision.

Delete “experiment”.

Authors’ response: [p.10 L327]

The term “experiments” has been modified to “measurements” in all instances.

Recommend deleting Table 4. These data are also shown in Figure 4, and the few significant correlations are also given in the text.

Authors’ response: [supplementary material]

Table 4 has been removed from the manuscript, and instead we have included it in the supplemental material as Table S3 as we believe that this data may be valuable to the journal’s readership. However, we are more than happy to reconsider this decision on the Editor’s recommendation.

Table 2. I don’t think it is necessary to show the regression statistics for all 48 SMLR models tested. It is sufficient to show the best model for each site.

Authors’ response: [p.11 L365]

In Table 2 in the revised manuscript, we have deleted all SMLR models except the best model for each site.

Some of the discussion of the PCA analysis doesn’t make sense to me, regarding which factors have the greatest effect. For example, for GCW lines 510-511 state that C.dominance and Q7day have the greatest effect on the 1st and 2nd axes. However, Table 7 appear to show higher contributions for other factors, including EC and SiO2 on axis 1, and Temp, TN, and NO3N on axis 2. Similarly for DSW, line 520 says C.dominance has greatest effect on the 1st and 2nd axes. However, in Table 7 several other factors have larger contributions on both axes

Authors’ response: [p.16 L493-495, p.17 L505-506, p.17 L511-512, p.17 L523-524]

We have revised the descriptions for the major factors that contribute on each principal component axes according to Table 4 (Table 7 in the original manuscript) and Figure 8.

 Figure 9, a few of the points are larger. Is there any meaning to that? They do not appear to be samples with particularly high cyanobacteria, since some of them have a normal HAB level. Also, the shaded areas appear to be intended to include all the samples with a particular HAB level – were they just drawn to illustrate that or was some statistical procedure used to develop them?

Authors’ response: [Figure 8]

We used PCA () function in FactoMineR R package for computing principal component analysis. The larger symbols indicate the group mean point for each HAB level. The shaded areas are drawn using fviz_pca_ind () function with the argument “addEllipses = TRUE”. We have added the meaning of larger symbols in the caption of Figure 8.

Inconsistency – Line 640 argues that TP is high enough that it doesn’t limit algal growth. However, DSW, the one site that TP appeared to be an important factor, had the highest mean TP concentration. I note that Table 1 indicates DSW also has the largest standard deviation for TP, so maybe that higher variability means that site oscillates between TP limitation and excess, while the other sites more consistently have excess TP?

Authors’ response: [p.10 L336-337]

As described on page 7 (Line 270-271), a contaminated tributary named Geumho River enters the upstream of DSW, which has a great effect on the variations of the TP concentrations in the DSW site. From this study, it is difficult to define the reason that TP limits algal growth only in the DSW, but the high TP variability could be one explanation. We appreciate the reviewer’s insightful comments on the TP limitation in the DSW. We added following description in the revised manuscript [p. 10 L 336-337].

“It is difficult to define why algal growth was TP-limited only at DSW, however the high TP variability (Table 1) could provide an explanation

Reviewer 4 Report

This is an interesting paper that utilises a moderate sized data set to examine the factors predicting or influencing cyanobacterial dominance. They utilise several statistical methods to determine the influencing factors. The subject matter is important but I think the manuscript needs more literature pertaining to rivers and weir pools and the present state of knowledge on this subject. It draws too widely from the “general” cyanobacterial literature where it should be focusing more on rivers and cyanobacterial blooms. This needs to be done in a revision. Also the manuscript is too long in general. Some of the statistics / model description is too long and I am not sure all the figure and tables are required. Below I give some suggestions for which I think could be removed.

The discussion is largely missing and the present work needs to be put in the context of cyanobacterial bloom research in rivers and weir pools. Papers missing that have also found stratification among other factors to be important include those by Bormans, Sherman, Mitrovic among others. The data collected here needs to be put in context of the past research on Rivers. Several paragraphs are needed for this.

Also some of the factors measured and analysed co-vary with others and some may not be as important to include in the analysis as they may not influence cyanobacteria greatly at the levels in the study. EC was one of the most important variables associated with cyanobacterial dominance. However it is unlikely to affect cyanobacteria greatly at the concentrations measured. As the authors suggest it is probably more related to low flows leading to increased EC. Why it is more predictive is not clear. Perhaps this means that deeper analysis of the data is required. For instance testing Q1, and Q14 to see if any of these are more related to cyanobacteria. I would suggest removing EC from the analysis, as well as Chl a which is not independent of cyanobacteria (cyano conc effect chl a).   

Some more specific comments

Line 46. Complex interaction

81-83 P release from the  sedimentary layer occurs when the oxygen (O) supply to the lower layer is limited in stratified and 82 stabilized water layers and the water becomes anaerobic; this can also promote cyanobacteria  proliferation [26,27]. – this should be more specific – talk about hypolimnial oxygen depletion and what causes it – lack of light, oxygen demand from the sediment etc.  

93. are known to be an important factor influencing cyanobacterial outbreaks [14−16]. Do you mean which species becomes dominant based on the rate of buoyancy?

94. Furthermore, cyanobacteria form communities and – I think you mean colonies? Should mention they also form filaments/trichomes and how both colony formation and trichomes impact grazing.

310 This behavior seems to have been because several algae classes grew in  competition with each other during the survey period, rather than any specific class achieving 311 dominance. Cite the Bormans / Sherman references for transitions between cyanobacteria and diatoms.

316-319. Furthermore, negative correlations were obtained between the Fe concentration and cyanobacteria, green algae, 317 and diatom cell densities at every weir (r = −0.34 to −0.07). Fe is known to accelerate the 318 photosynthesis of cyanobacteria; however, no significant correlations were found in the correlation 319 analysis. – Need a reference for this, and perhaps the lack of “positive” correlations was that the cyanobacteria were not Fe limited?

327-328. The correlation between the algae biomass (Chl-a) and nutrients in the study weirs are shown 328 in Figure 4 and Table 4. I don’t think that Figure 4 adds much more than the table does and suggest removing Figure 4.

329. The experiment results showed… Its not really an experiment but monitoring. This should be changed throughout the manuscript.

387- 388  RFE is a backward method for selecting a parsimonious model that achieves the maximum prediction performance using the minimum variables, in which variables with low importance are  removed one by one.  - This reads like it should be in the methods / statistics section. Also do you mean “backward selection method”?

Note that EC features among the environmental conditions for high cyanobacterial shares for 466 each weir because this property is susceptible to flow rate variations, and not because it promotes 467 cyanobacterial growth (Figure 8). As shown in Figure 8, EC tended to increase as the river flow 468 decreased at each weir. Thus, EC can be considered an indirect index indicating a continuous low 469 flow condition in the river. In other words, when drought persists and the natural runoff quantity 470 due to rainfall-runoff is insufficient, EC increases. This is because the river water quality is greatly 471 affected by the groundwater and effluent from large sewage treatment plants. Furthermore, EC 472 may increase due to anaerobicization of the water-sedimentary interface due to thermal 473 stratification and the subsequent release of ionic materials [63,64].

This paragraph is important and to raises the question of whether EC should be included in the model. If EC is not directly influencing cyanobacterial growth (as I believe it is not) and is just a product of the lower flows, it might be better to remove it from the analysis. The results currently suggest EC as being an important predictor.     The last sentence is also important, and if any DO profile data is available this would help to distinguish periods when thermal stratification led to reduced hypolimnial oxygen levels.  

Author Response

Thank you very much for your careful review and suggestions that have helped us improve the quality of our manuscript. Reviewers’ comments/suggestions are in bold text. Author responses are in plain text. All revisions are highlighted in the revised manuscript.

Response to Reviewer #4

The manuscript needs more literature pertaining to rivers and weir pools and the present state of knowledge on this subject

Authors’ response: [p.1 L40-43, p. 2 L44-45, p. 17 L502-504]

We agree that this information was lacking, and have added references [9, 10, 11, 12, 13] to the existing references [21, 22, 53, 63, 70, 72] pertaining to rivers and weir pools.

The manuscript is too long in general

Authors’ response:

We moved Tables 2, 3, 4 and Figure 6 from the original manuscript to the supplemental material. In Table 2 in the revised manuscript, we have also deleted all SMLR models except the best model for each site. We also removed some sentences in the introduction to focus on the rivers and weir pools. The length of the manuscript has been reduced from 32 pages to 25 pages.

Papers missing that have also found stratification among other factors to be important include those by Bormans, Sherman, Mitrovic among others.

Authors’ response: [p.1 L40-43, p. 2 L44-45, p. 17 L502-504]

We have included the suggested papers by Bormans [11], Sherman [10], and Mitrovic [13] and cited these references in the introduction and the results and provided comparisons with our findings.

“The results are in a good agreement with the findings in the lowland rivers of south-eastern Australia [10-11,13], where a persistent low flow and formation of thermal stratification caused proliferation of cyanobacteria.” [p. 17 L502-504]

As the authors suggest it is probably more related to low flows leading to increased EC. Why it is more predictive is not clear. Perhaps this means that deeper analysis of the data is required. For instance testing Q1, and Q14 to see if any of these are more related to cyanobacteria. I would suggest removing EC from the analysis, as well as Chl a which is not independent of cyanobacteria (cyano conc effect chl a).

Authors’ response: [p.13 L449-455]

We have included EC because it is a good indicator of river water pollution by point sources during low flow periods. Furthermore, as it is easy to measure in-situ with sensors, EC can be used as the most important variable for prediction of algal blooms, if high-frequency measurement data can be secured in real time together with water temperature data.

We tested Q1, Q3, and Q7 during the study, and determined Q7 as the best predictor in the site.

The correlation coefficients between Chl-a and cyanobacteria are different for each site [Table S2 and S3]. The relationship is highly variable for different season either depending on the dominant phytoplankton classes. For instance, the correlation coefficient in the CHW site is just 0.04, while that for the DSW site is 0.99. This is because the total Chl-a is dependent on all algae species (not only on cyanobacteria) and the content of Chl-a per cyanobacteria cell is lower than green algae and diatoms. A recent study (Khac et al., 2018, Processes 6) showed that the ratio between phycocyanin to Chl-a concentrations can be a good indicator of the development of algal bloom by dominance of cyanobacteria.

Complex interaction

Authors’ response: [p.2 L47]

We have corrected it.

 [Line 76-78] “P release from the sedimentary layer occurs when the oxygen (O) supply to the lower layer is limited in stratified and stabilized water layers and the water becomes anaerobic; this can also promote cyanobacteria proliferation [26,27].” – this should be more specific – talk about hypolimnial oxygen depletion and what causes it – lack of light, oxygen demand from the sediment etc.

Authors’ response: [L73-75]

 We have modified the sentence to make it more specific as follows.

“Release of P from the sedimentary layer occurs when the oxygen (O) supply to the lower layer is limited and sediment oxygen demand is high in stratified water layers causing the water to become anaerobic”

[Line 88] “ are known to be an important factor influencing cyanobacterial outbreaks [1416]”. Do you mean which species becomes dominant based on the rate of buoyancy?

Authors’ response: [p.2 L84-85]

To clarify our description, we have changed it to “Buoyancy mechanisms of cyanobacterial species such as Microcystis and Oscillatoria are important factors influencing cyanobacterial outbreaks [36-38]”.

[Line 86] “Furthermore, cyanobacteria form communities and” – I think you mean colonies? Should mention they also form filaments/trichomes and how both colony formation and trichomes impact grazing.

Authors’ response: [p.2 L86-87]

To clarify our description, we have changed it to “Furthermore, cyanobacteria form spherical colonies, filaments, and trichomes surrounded by mucus and have less zooplankton predation than other algae [35,39-40].”

[Line305] “This behavior seems to have been because several algae classes grew in competition with each other during the survey period, rather than any specific class achieving dominance.” Cite the Bormans / Sherman references for transitions between cyanobacteria and diatoms

Authors’ response: [p.8 L308-309]

We cited Bormans [11] and Sherman [10] to support the statement.

 [Line 310-312] “Furthermore, negative correlations were obtained between the Fe concentration and cyanobacteria, green algae, and diatom cell densities at every weir (r = −0.34 to −0.07). Fe is known to accelerate the photosynthesis of cyanobacteria; however, no significant correlations were found in the correlation analysis.” – Need a reference for this, and perhaps the lack of “positive” correlations was that the cyanobacteria were not Fe limited?

Authors’ response: [p.9 L 317-318]

We added a reference [66] for the sentence. We note that total iron in freshwaters, in general, is about 10-7-10-5 M Fe, but most is present in flocs and particulates (Ecology of Phytoplankton, Reynolds, 2006) with extremely low dissolved concentrations (                                               10-15 M Fe). However, the Fe concentrations measured in the river water at the study sites were relatively high, i.e., in the ranges of 10-6-10-7 M Fe. Therefore, as for SiO2, the lack of positive correlations may be because of the abundance of Fe.

 [Line 321-322] The correlation between the algae biomass (Chl-a) and nutrients in the study weirs are shown in Figure 4 and Table 4. I don’t think that Figure 4 adds much more than the table does and suggest removing Figure 4.

Authors’ response: [supplemental material]

Table 4 has been removed from the manuscript, and is now included in the supplemental material as Table S3.

 [Line 329] “The experiment results showed…” Its not really an experiment but monitoring. This should be changed throughout the manuscript.

Authors’ response: [p.1 L13, p.3 L121, p.3. L127, p.5. L183-184, p.10. L327]

The reviewer is correct. All references to “experiments” used in the manuscript have been amended to “measurements”. To clarify, all water quality and algae data, except for the meteorological and flow data, used for statistical analysis were collected by the authors through field monitoring and laboratory measurements.

 [Line 218- 220] “RFE is a backward method for selecting a parsimonious model that achieves the maximum prediction performance using the minimum variables, in which variables with low importance are removed one by one.” – This reads like it should be in the methods / statistics section. Also do you mean “backward selection method”?

Authors’ response: [p.6 L212-214]

We agree with the reviewer and appreciate this advice. The sentence has been moved to section 2.3 Statistical Analyses, and “backward method” has been changed to “backward selection method”.

 “Note that EC features among the environmental conditions for high cyanobacterial shares for each weir because this property is susceptible to flow rate variations, and not because it promotes cyanobacterial growth (Figure 8). As shown in Figure 8, EC tended to increase as the river flow decreased at each weir. Thus, EC can be considered an indirect index indicating a continuous low flow condition in the river. In other words, when drought persists and the natural runoff quantity due to rainfall-runoff is insufficient, EC increases. This is because the river water quality is greatly affected by the groundwater and effluent from large sewage treatment plants. Furthermore, EC may increase due to anaerobicization of the water-sedimentary interface due to thermal stratification and the subsequent release of ionic materials [62,63].”

This paragraph is important and to raises the question of whether EC should be included in the model. If EC is not directly influencing cyanobacterial growth (as I believe it is not) and is just a product of the lower flows, it might be better to remove it from the analysis. The results currently suggest EC as being an important predictor. The last sentence is also important, and if any DO profile data is available this would help to distinguish periods when thermal stratification led to reduced hypolimnial oxygen levels.

Authors’ response: [p.19 L556-566]

While we understand the Reviewer’s concern regarding the influence of EC, after much consideration we believe that this information is valuable in the manuscript (please see our previous detailed comments in this regard in response to Reviewer comment (4).

Round 2

Reviewer 4 Report

The authors have adequately addressed my concerns. It is now suitable for publishing.